# Replicable simulation of distal hot water premise plumbing using convectively-mixed pipe reactors

**M. Storme Spencer, Abraham C. Cullom, William J. Rhoads, Amy Pruden◉, Marc A. Edwards***

Department of Civil and Environmental Engineering, Virginia Tech, Blacksburg, VA, United States of America

* edwardsm@vt.edu

**Data Availability Statement:** Sequence data have been deposited in the NCBI Sequence Read Archive (BioProject ID PRJNA609991). Additional relevant

## Abstract

A lack of replicable test systems that realistically simulate hot water premise plumbing conditions at the laboratory-scale is an obstacle to identifying key factors that support growth of opportunistic pathogens (OPs) and opportunities to stem disease transmission. Here we developed the convectively-mixed pipe reactor (CMPR) as a simple reproducible system, consisting of off-the-shelf plumbing materials, that self-mixes through natural convective currents and enables testing of multiple, replicated, and realistic premise plumbing conditions in parallel. A 10-week validation study was conducted, comparing three pipe materials (PVC, PVC-copper, and PVC-iron; n = 18 each) to stagnant control pipes without convective mixing (n = 3 each). Replicate CMPRs were found to yield consistent water chemistry as a function of pipe material, with differences becoming less discernable by week 9. Temperature, an overarching factor known to control OP growth, was consistently maintained across all 54 CMPRs, with a coefficient of variation <2%. Dissolved oxygen (DO) remained lower in PVC-iron (1.96 ± 0.29 mg/L) than in PVC (5.71 ± 0.22 mg/L) or PVC-copper (5.90 ± 0.38 mg/L) CMPRs as expected due to corrosion. Further, DO in PVC-iron CMPRs was 33% of that observed in corresponding stagnant pipes (6.03 ± 0.33 mg/L), demonstrating the important role of internal convective mixing in stimulating corrosion and microbiological respiration. 16S rRNA gene amplicon sequencing indicated that both bulk water ($P_{adonis}$ = 0.001, $R^2$ = 0.222, $P_{betadis}$ = 0.785) and biofilm ($P_{adonis}$ = 0.001, $R^2$ = 0.119, $P_{betadis}$ = 0.827) microbial communities differed between CMPR versus stagnant pipes, consistent with creation of a distinct ecological niche. Overall, CMPRs can provide a more realistic simulation of certain aspects of premise plumbing than reactors commonly applied in prior research, at a fraction of the cost, space, and water demand of large pilot-scale rigs.

## Introduction

Opportunistic pathogens (OPs), such as *Legionella*, non-tuberculous mycobacteria, and *Pseudomonas aeruginosa*, now account for the primary source of tap-water associated disease in the U.S. and much of the world [1, 2]. Because these organisms grow in the premise (i.e.,

data are within the manuscript and its Supporting Information files.

**Funding:** This work was supported by the National Science Foundation (CBET award number 1706733, nsf.gov) and the Center for Disease Control and Prevention (contract number 75D30118C02905, cdc.gov), and a National Science Foundation Graduate Fellowship to Abraham Cullom. The funders had no role in study design, data collection and analysis, decision to publish, or preparation of the manuscript.

**Competing interests:** The authors have declared that no competing interests exist.

building) plumbing environment, rather than originating from fecal-contaminated source water, substantial attention has been expended towards identifying factors that stimulate their growth [3]. However, such efforts are hampered by lack of a suitable laboratory test apparatus that accurately represents premise plumbing conditions, while also being replicable to provide sufficient statistical power for evaluating various factors of interest [4].

There are numerous challenges to effectively simulating premise plumbing conditions in the laboratory. Existing approaches inevitably involve compromise in terms of complexity, cost, replicability, and/or ability to achieve relevant hydraulic and water chemistry regimes. Annular reactors and CDC biofilm reactors [5, 6] are designed with the intention of enabling continuous or semi-continuous flow with coupons spinning internally to produce hydraulic sheer stress, such as that experienced by a biofilm on the interior pipe wall [4, 7, 8]. However, these and other bench-scale plumbing simulations still do not achieve realistic plumbing flow patterns, hinder replication through their large size and cost, and are characterized by large amounts of unrepresentative surface area comprised of materials not used in plumbing including glass and plastic. More realistic pilot-scale plumbing simulations are large, costly, and require very large volumes/flows of water, which also makes influent water chemistry conditions difficult to precisely control [9, 10]. Pilot-scale studies examining OPs also typically require direct connection to premise plumbing of the study facility and cannot be sampled within the protection of a biological safety-level (BSL) 2 certified cabinet, elevating potential for exposure of workers to pathogen-containing aerosols during sampling and thus requiring appropriate institutional approvals.

Hydraulic conditions create distinct ecological niches, e.g., with mostly stagnant versus continuously flowing pipes representing two extremes, by controlling the temperature and delivery of disinfectants (e.g., chloramine, chlorine, copper) or nutrients (e.g., oxygen, organic carbon, nitrogen, phosphorus) to biofilms [11, 12]. Alternating periods of flow and stagnation create extreme fluctuations in temperature, disinfectant concentrations, nutrients, and metabolic products in the water that control microbial growth rates and can select for certain organisms [13]. Field measurements of water drawn from premise plumbing following overnight stagnation have documented over 3-log increases in total heterotrophic bacterial counts as a result of depleted disinfectant residuals [14, 15].

On the other hand, recent research has revealed that portions of premise plumbing from which consumers are not drawing water are not truly stagnant during periods of non-use, but can be subject to rapid internal convective mixing due to temperature gradients [16]. Convective mixing is characteristic of certain distal reaches of hot water premise plumbing, resulting in gentle circulation and sustained warm temperatures known to be ideal for growth of *L. pneumophila* and other OPs. Orientations in which hot water flows upwards and cools drives convective mixing and has correspondingly been observed to affect microbial community composition [17]. In another illuminating experiment, intermittent delivery of hot water at 51˚C followed by rapid cooling to room temperature during stagnation produced two orders of magnitude higher levels of *L. pneumophila* compared to a situation with more frequent flow events delivering 51˚C [10]. This is to be expected based on the common practice of heating water samples prior to culture in order to select for heat-tolerant *Legionella* versus conventional heterotrophs.

Here we introduce the convectively-mixed pipe reactor (CMPR) as a simple, replicable, and realistic system for premise plumbing simulations to evaluate factors contributing to the growth of OPs. The CMPR consists of off-the-shelf materials used in real-world plumbing systems. Capped pipe segments, with one end submerged in a hot water bath and the other contacting the cooler ambient air, simulate the premise plumbing riser from a hot water recirculation loop that connects to distal, stagnant outlets (Fig 1A). This temperature gradient

## A – Schematic of CMPR Configuration

## B – Internal Convective mixing in a CMPR

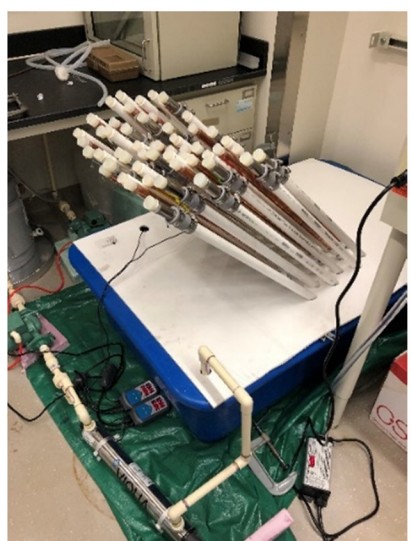
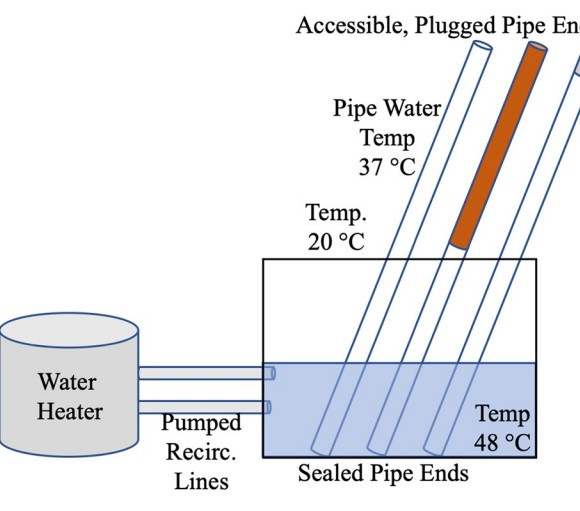
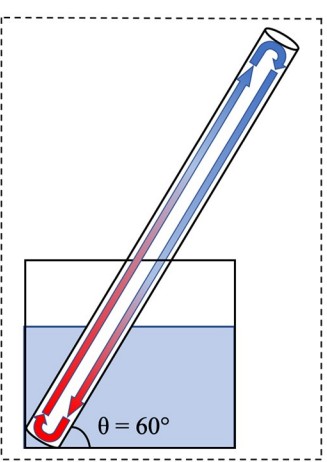

**Fig 1. Schematic of convective mixing pipe reactors (CMPRs).** (A) Capped ends of PVC-copper, PVC-iron, and PVC pipes are submerged at 60° in a hot water bath to simulate hot water recirculation, with plugged, accessible ends exposed to room temperature to simulate a stagnant distal outlet. Fifty-four pipes, configured into 6 rows of 9 pipes, were operated for this study. In-line ultraviolet light is incorporated for disinfection of water bath for secondary containment/disinfection of pathogens in case of leak to support BSL2 level experiments. (B) This configuration of the pipes induces convective mixing in the interior bulk water.

recreates natural internal flow via convective currents (Fig 1B), constantly circulating water without use of pumps or mechanical mixers, substantially reducing the cost, maintenance requirements, and issues associated with inevitable mechanical malfunction. The simple design supports closed system operation of replicates to enable statistical rigor in experimental design. This study evaluates the overall reproducibility of physicochemical properties and microbial community compositions produced by CMPRs using three pipe materials (PVC, PVC-copper, and PVC-iron) as compared to the same configurations maintained under stagnant, constant temperature conditions. The findings are put into context with other studies to evaluate relative advantages and disadvantages of alternative premise plumbing simulations.

## Materials and methods

### Convectively mixed pipe reactor (CMPR) design and operation

Passive, convective mixing was achieved by submerging the capped end of sealed 4-ft pipe segments (CMPRs) into a heated water bath housing unit at a 60° angle with the other half exposed to the ambient room temperature (Fig 1A). A 60° angle was chosen because it maximizes convective mixing (Fig 1B, S1 Fig and S1 Table). An internal circulating water velocity of ~40 cm/s was determined by injecting neutrally buoyant Rhodamine dye pre-heated to 37°C into a clear PVC CMPR and timing the mixing velocity via video and stopwatch. Each end of the CMPRs were sealed with silicone stoppers to prevent changes in pH associated with atmospheric $CO_2$ transfer.

The CMPRs were composed either entirely of ¾" PVC (Silver-line Plastics, Ashville, NC) (PVC pipes/CMPRs), half ¾" PVC and half type M copper pipe (McMaster-Carr, Elmhurst, IL) (PVC-copper pipes/CMPRs), or 3" of mild steel pipe (McMaster-Carr, Elmhurst, IL) attached to 45" of ¾" PVC (PVC-iron pipes/CMPRs). These three materials exhibited distinct heat conduction properties and would in turn create distinct convective mixing patterns and

velocities if submerged in the water bath. Thus, for purposes of this experiment, PVC segments were submerged in the water bath to normalize convective mixing across the conditions. To ascertain a high-resolution view of the variability of chemical and biological water quality parameters among CMPRs, 18 replicates of each material were used. In addition, triplicates of each pipe were placed at a 60˚ angle against a shelf in a constant temperature room averaging 38˚C (ideal for growth of *Legionella* and other OPs) to compare the effects of convective mixing to fully stagnant, heated pipes.

The influent water to both CMPRs and stagnant pipes was initially seeded with backwash water from a granular activated carbon filter that had been in operation in a premise plumbing drinking water system for > 2 years to establish a mature microbial community. Water changes were conducted once weekly for the first 2 weeks of operation to facilitate colonization of the pipe surfaces. Water changes were then increased to twice weekly to better simulate an infrequent use pattern characteristic of distal outlets in large buildings, as a possible worst-case scenario for OPs control, until the end of the experiment (week 10). These periodic manual dump and fill water changes served to recreate turbulent intermittent flow and complete changeover of water, as occurs at infrequently used distal outlets. In order to mitigate changes in pH due to headspace and exchange with the atmosphere, and to prevent spills to facilitate testing of pathogens, equal volumes of water were measured and poured into the pipes to maintain consistent headspace (~1") between CMPRs. Influent water was prepared with de-chloraminated Blacksburg, VA tap water via breakpoint chlorination to destroy residual ammonia. The remaining chlorine residual was quenched with sodium thiosulfate, and pH was adjusted to 7.5 using 1 M hydrochloric acid. A summary of influent water characteristics is provided in S2 Table.

The water bath in which the CMPRs were partly immersed was heated by a 22.7 L (6 gal) water heater and continuously recirculated through the housing unit (a 48" x 30" x 15", 307 L (81 gal) fiberglass tank) to achieve a steady-state temperature of approximately 45–48˚C (which is typical of lower end of water heater temperature settings) as well as an in-line ultraviolet disinfection unit (Viqua, Guelph, ON) to prevent microbial proliferation in the water bath and for biosafety in the event of a spill, as well as achieve the target bulk water temperature in the CMPRs. One recirculation pump delivered water from the heater to the CMPR housing unit and another pump delivered water from the CMPR housing unit back into the water heater. The unit is enclosed with a high-temperature, ultra-high molecular weight plastic cover (McMaster-Carr, Elmhurst, IL) reinforced with steel rods and fixed in position by clamps. Water was supplied to the heater using a valve connection to periodically change and/or refill the housing unit.

## Water quality analysis

Temperature, dissolved oxygen (DO), pH, total organic carbon (TOC), and total and dissolved metals were measured at weeks 2 and 9 of the experiment. These timepoints were selected to profile the pipes following initial microbial colonization and at a later period when differences in biological water quality parameters would be expected based on pipe material [18, 19]. Samples were collected by inverting the pipe three times and decanting the contents into a sterile 1 L polypropylene bottle. pH and temperature were measured using a pH 110 meter with automatic temperature correction (Oakton Research, Vernon Hills, Il). DO was measured using a polarized DO probe (Thermo Fisher Scientific Orion 3-star meter, Waltham, MA). TOC was measured using a Sievers Model 5300C autosampler according to Standard Method 5310 C. Total and soluble copper (MDL = 0.08 ppb, MRL = 1.00 ppb) and iron (MDL = 0.36 ppb, MRL = 5.00 ppb) were measured following acidification with 2% v/v nitric acid using inductively coupled plasma mass spectroscopy (iCAP RQ ICP-MS; Thermo Fisher Scientific,

Waltham, MA). Samples for soluble metal analysis were immediately filtered through a 0.45-μm nylon filter (Whatman, Maidstone, UK) prior to acidification and measurement.

## Biological sampling

Bulk water total cell counts were taken during weeks 1, 2, 5, and 9, with an additional sampling of a random subset of CMPRs and pipes during week 6. Counts were measured using quantitative flow cytometry (BDAccuri C6; BD Biosciences, San Jose, CA) following staining with SYBR Green I fluorescent nucleic acid stain using previously developed methods [20, 21].

After 10 weeks, one pipe volume (~400 mL) was collected in a sterile, 1 L, polypropylene bottle following three inversions of the pipe for mixing, and then filter concentrated onto 0.22-μm mixed nitrocellulose ester membranes (Millipore, Billerica, MA). PVC caps on the ends of the pipes were removed using pipe cutters and swabbed and the entire inner surfaces (~13 cm$^2$) of these endcaps were swabbed with a sterile, cotton-tip applicator (Puritan, Guilford, ME) for biofilm sample collection in a single circular motion, making contact with the full length of the cotton portion of the applicator. The PVC cap surfaces were selected for swabbing to ensure a consistent recovery of biofilm-associated microbes across conditions, as corrosion/deposition phenomena on the metal surfaces were anticipated to interfere with biofilm recovery. DNA was extracted from fragmented filters or swab tips using a FastDNA Spin Kit (MP Biomedicals, Solon, OH) following manufacturer's protocols. Filter, swab, lab (autoclaved DI water exposed to laboratory conditions), and extraction blanks were included with sample processing. Quantitative polymerase chain reaction (qPCR) targeting the bacterial 16S rRNA gene to quantify total bacteria DNA was performed as described previously [22, 23]. Briefly, qPCR reactions were carried out in triplicate 10 μL reactions containing 1x SsoFast Evagreen Supermix (Bio-Rad, Hercules, CA), 400 nM of forward and reverse primers, and UV sterilized, molecular grade water with 1 μL of DNA template. Samples were diluted 1:10 to minimize inhibition and a standard curve was generated for each run using 10-fold serial dilutions of custom gBlock (Integrated DNA Technologies, Coralville, IA) gene fragments with a quantification limit of 1000 gene copies per reaction. Samples were considered quantifiable if at least 2 of 3 triplicates were above the limit of quantification.

## 16S rRNA amplicon sequencing

Sample preparation for 16S rRNA amplicon sequencing was performed following the Earth Microbiome Project protocol for amplification of the V4-V5 region of the 16S rRNA gene using the 515F/926R primer pair. Combined triplicate PCR products were pooled to 240 ng each and purified using the QIAquick PCR purification kit (Qiagen, Valencia, CA). Sequencing was performed on the Illumina MiSeq platform by the Biocomplexity Institute at Virginia Tech (300-bp, paired end reads). Sequencing reads were processed using the QIIME2 pipeline (v. 2019.1) [24]. Sequences were quality filtered and dereplicated using the DADA2 pipeline with forward reads truncated at 297 bp and reverse reads truncated at 200 bp [25]. Resulting amplicon sequence variants (ASVs) were maintained for downstream analyses. Remaining sequences were taxonomically classified using the Scikit-learn classifier [26] and a pre-trained Silva 16S rRNA (release 132) database [27] for 99% similarity (515F/926R region, seven-level taxonomy). Singleton ASVs were removed and taxonomy-based filtering to exclude ASVs identified as either mitochondria or chloroplast was performed. A total of 4,631,690 sequencing reads were maintained across the 137 samples with a mean of 33,807 reads, a minimum of 1,139 reads, and a maximum of 55,749 reads/sample. Sample reads were rarefied to 10,177 randomly-selected reads using the phyloseq package (v.1.28.0 [28]), which excluded 11 samples below the threshold read count (lab blank, extraction blank, a filter blank, and 8 samples).

Influent water samples, seeded influent water samples, along with filter, swab, and extraction blanks were included in the sequencing lane. Sequence data have been deposited in the National Center for Biotechnology Information (NCBI) Sequence Read Archive (SRA) under BioProject ID PRJNA609991.

## Statistical analyses

Statistical analyses of chemical and biological parameters were performed in JMP Pro 14 using the Wilcoxon test for comparing two groups or the Kruskal-Wallis test for comparing more than two groups, followed by the Dunn test for multiple comparisons. Statistical testing for microbial community data was performed in R Studio (v.1.2.1335) using R version 3.6.0. Alpha diversity in microbial community data was carried out with the phyloseq package (v.1.28.0 [28]) using the pairwise wilcoxon test (pairwise.wilcox.test) using a Benjamini-Hochberg procedure. Beta diversity was analyzed by applying the permutational multivariate analysis of variance (Adonis) [29] found in the vegan package (v.2.5.5) [30] to the Bray-Curtis dissimilarity matrix generated using the phyloseq package. Multivariate homogeneity of group dispersions analysis (betadisper) [31] was also applied to check the homogeneity of dispersion among groups assumption for Adonis. Non-metric multi-dimensional scaling (NMDS) using the Bray-Curtis Dissimilarity matrix was generated using the phyloseq package to ordinate data for comparison. Differential abundance of ASVs was determined utilizing DESeq2 using a Wald test for significance (p<0.01) [32].

## Results and discussion

### Trends and variance among the CMPRs

**Physicochemical consistency among the CMPRs.** *Temperature.* was highly consistent across CMPRs, as indicated by a coefficient of variation (CV) for each pipe type of <2% for both weeks 2 and 9 (Table 1). The experimental design successfully achieved ideal OP growth temperatures across all pipe types, with 37.6 ± 0.7°C during week 2 and 39.6 ± 0.6°C during week 9 (Fig 2D). Temperature is considered to be an overarching factor impacting OP growth and proliferation in premise plumbing systems [3, 33, 34] ().

**Table 1. Water quality parameters for replicate CMPRs during week 2 (acclimation, once weekly water change), week 9 (differentiation, twice weekly water change) or week 10 (qPCR data).**

|  | Week 2 | | | Week 9/10 | | |
|---|---|---|---|---|---|---|
|  | **Copper** | **Iron** | **PVC** | **Copper** | **Iron** | **PVC** |
| *Physicochemical Parameters* | | | | | | |
| (% Coefficient of Variation, 100 * Standard Deviation / Mean) | | | | | | |
| **Temperature (°C)** | 1.71% | 1.64% | 1.92% | 1.29% | 1.51% | 1.72% |
| **pH** | 2.15% | 2.96% | 0.88% | 2.30% | 2.80% | 0.04% |
| **DO (mg/L)** | 2.60% | 11.60% | 12.00% | 6.36% | 14.70% | 3.85% |
| **TOC (mg/L)** | 5.26% | 7.54% | 28.50% | 11.70% | 30.20% | 8.86% |
| **Total Cu (mg/L)** | 32.20% | - | - | 30.40% | - | - |
| **Soluble Cu (mg/L)** | 21.50% | - | - | 35.40% | - | - |
| **Total Fe (mg/L)** | - | 18.80% | - | - | 14.90% | - |
| **Soluble Fe (µg/L)** | - | 361% | - | - | 232% | - |
| *Biological Parameters* | | | | | | |
| **Total Cell Counts (events/µL)** | 231% | 43.80% | 17.80% | 25.40% | 26.40% | 18.20% |
| **Bulk Water 16S rRNA (log [gc/mL])** | - | - | - | 2.75% | 6.33% | 1.58% |
| **Biofilm 16S rRNA (log[gc/cm$^2$])** | - | - | - | 7.02% | 6.33% | 11.60% |

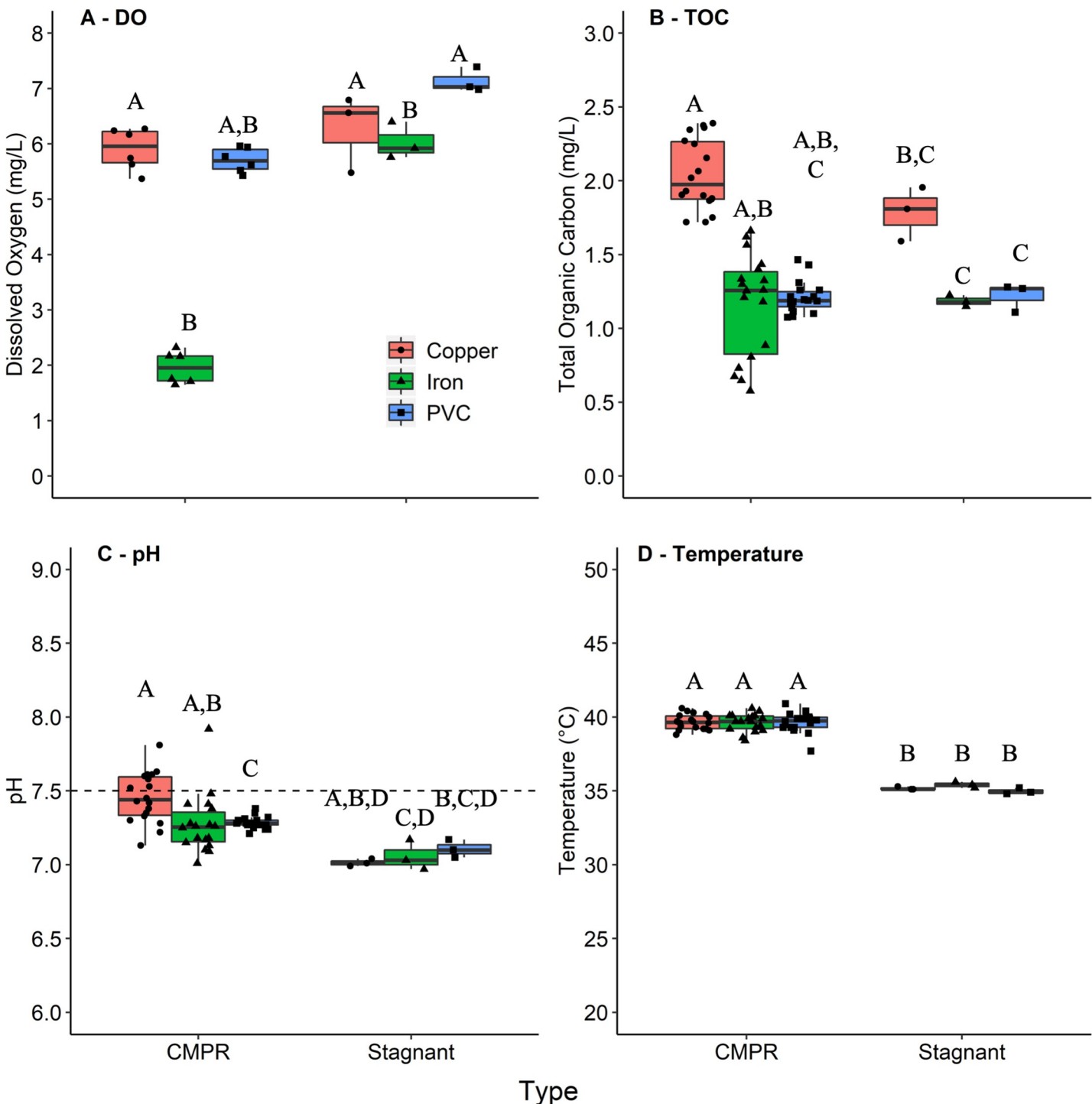

**Fig 2. Physicochemical comparison between CMPRs and stagnant pipes.** Comparison of (A) DO, (B) TOC, (C) pH, and (D) temperature among CMPRs (random subset of n = 6 for DO, n = 18 for all others) and stagnant incubator room pipes (n = 3) after 9 weeks of aging. Matching letters indicate groupings based on Dunn's test results (p < 0.05). Statistical groupings are independent for each panel.

**pH** was consistent with known trends for each type of pipe and reproducible. The targeted influent pH of 7.50 ± 0.05 was achieved. In week 2, PVC-copper pipes had a final pH of 7.66 ± 0.16 and PVC-iron pipes 7.72 ± 0.23, with a lower pH of 6.85 ± 0.06 in the PVC pipes

($n_{each}$ = 18, p<0.0001 compared to each PVC-copper and PVC-iron CMPRs), presumably due to increased $CO_2$ production via cellular growth and respiration [35, 36] in PVC CMPRs and corrosion in metallic CMPRs [37, 38]. An increase in bulk water pH from the service line to outlets has been previously observed in a residential building featuring copper plumbing [39]. The CV remained low in all pipes through week 9 (Table 1), when the difference in pH as a function of pipe type decreased, with average pH of 7.45 ± 0.17 in PVC-copper, 7.27 ± 0.20 in PVC-iron, and 7.28 ± 0.04 in PVC (Fig 2C). PVC-copper pipes had a slightly higher pH on average than both PVC ($n_{each}$ = 18, p = 0.0115) and PVC-iron ($n_{each}$ = 18, p = 0.0006). pH is important, as it can strongly shape microbial community composition [40] and also influence key aspects of water quality affecting OP proliferation, especially disinfectant efficacy [41–43] and rate of release of pipe corrosion products, which can either be bactericidal and/or serve as micronutrients [21].

*DO*. also shifted during stagnation as a function of pipe material. DO in the PVC-iron pipes (1.96 ± 0.29 mg/L) was 34% that of PVC (5.71 ± 0.22 mg/L, $n_{each}$ = 6, p = 0.0331) and 33% that of PVC-copper (5.90 ± 0.38 mg/L, $n_{each}$ = 6, p = 0.0043) pipes in week 9 samples (Fig 2A). This is expected due to corrosion of iron pipes consuming oxygen over time [44–46]. DO tended to be lower in the PVC pipes than in the PVC-copper pipes during week 2 ($n_{copper}$ = 9, $n_{PVC}$ = 6, p = 0.0018), although there was no difference between PVC-copper and PVC pipes following 9 weeks of acclimation. This is likely due to the aging of PVC-copper pipes, resulting in increased cell growth due to the decreased release of antimicrobial copper as the pipes aged [19, 47], which in turn allowed more cellular respiration to consume DO, as was observed earlier in the experiment for the PVC pipes.

*TOC*. in the influent averaged 0.81 ± 0.12 ppm and increased following incubation in the CMPRs. Week 2 effluent TOC was fairly high; averaging 6.99 ± 0.37 mg/L in PVC-copper, 4.03 ± 0.30 mg/L in PVC-iron, and 3.68 ± 1.05 mg/L in PVC pipes. After 9 weeks, the TOC decreased and stabilized at 2.04 ± 0.24 mg/L in PVC-copper, 1.16 ± 0.35 mg/L in PVC-iron, and 1.21 ± 0.11 mg/L in PVC pipes (Fig 2B). This decrease from week 2 to week 9 was likely due to washout of TOC leaching from new materials. The PVC-copper condition in particular required extra epoxy to join the PVC and copper pipe junctions. The largest variability in TOC was observed in the PVC-iron pipes during week 9, which also had the highest variability in DO with a CV of 14.7%; however, the standard deviation was low in magnitude (0.35 mg/L). This indicates that pipes aged and formed scale with time as occurs in real-world plumbing.

*Total copper and iron*. concentrations in the respective CMPRs also decreased with pipe aging (Table 1). Consistent with typical levels of copper in new pipes with time [48], average total copper levels were initially 0.66 ± 0.21 mg/L after 2 weeks of aging, but decreased to 0.35 ± 0.10 mg/L by week 9. Variability in copper release was fairly high, as is characteristic in new pipes [49, 50], with CVs of 32.2% during week 2 and 30.4% during week 9. This could be due to differential aging rates in individual pipes or differential release of pipe scale during the sampling procedure. Average soluble copper was stable at 0.23 ± 0.05 mg/L at week 2 and 0.29 ± 0.10 mg/L at week 9. Average total iron decreased from 24.9 ± 4.68 mg/L at week 2 to 10.9 ± 1.62 mg/L at week 9, consistent with scale formation [51]. The variability in iron release was expected based on previous studies [46, 52, 53]. Between weeks 2 and 9, the CV decreased from 18.8% to 14.9% across CMPRs, consistent with iron release becoming more uniform as pipe-scale formed. The average soluble iron was 110 µg/L at week 2 and 90 µg/L at week 9, indicating that the vast majority of iron in the bulk water was in particulate form (>99%).

**Microbiological characteristics of the CMPRs.** Microbiological profiling indicated somewhat greater variability among replicate CMPRs than the physicochemical parameters, particularly for the metal pipe materials (Table 1). PVC was the least variable pipe type with regard to total cell counts, for both week 2 (CV = 17.8%) and week 9 (CV = 18.6%). Total cell

counts were initially widely variable for PVC-copper pipes (CV = 231%), possibly due to different aging rates and toxicity impacts of variable copper release rates (CV = 32.2%). More variability was also observed in PVC-iron pipes relative to PVC (CV = 43.8%), which could also be due to differential rates of initial aging in the pipes given that pipes were new at the outset of the experiment (soluble iron CV = 361%). However, by week 9 the variability in total cell counts in PVC-copper CMPRs decreased and became similar to the variability displayed by PVC-iron pipes, despite soluble iron remaining variable (CV = 232%). We hypothesize that the greater initial variability in PVC-copper pipes was due to cupric ions being released at different rates, resulting in different rates of microbial inactivation. In PVC-iron pipes, high DO consumption by corrosion may explain the initial variability of total cell counts (S2 Fig).

*Total bacteria measured by qPCR.* targeting the 16S rRNA gene in the bulk water during week 10 of the experiment (Fig 3A) was less variable than total cell count metrics measured during week 9 (CV = 1.58–6.33%) (Table 1). The only pipe type with a range of variability more than one order of magnitude were PVC-iron CMPRs, with a range of 1.24 log(gc/mL). This could be related to the heterogeneous water chemistry produced by the iron condition due to insoluble iron oxide formation.

Biofilm total bacterial density, as estimated by 16S rRNA gene copy numbers, followed a similar trend as that in the bulk water, except there was somewhat more variance in the PVC-copper and PVC conditions (7.02% and 11.6% respectively). Similar levels of 16S rRNA genes were observed across the pipe materials in the biofilm at week 10 (Fig 3B; Table 1).

**Effect of pipe type on bacterial growth with convective mixing.** Total cell counts were measured on a weekly basis to track expected changes in microbial numbers with pipe age (Fig 4A) [18, 54–56]. During week 1, pipe type had a pronounced effect, with total cell counts in PVC-copper CMPRs being lower than in PVC and PVC-iron CMPRs ($n_{each}$ = 18, p<0.0001 and p = 0.0015, respectively) and PVC being higher than PVC-iron ($n_{each}$ = 18, p = 0.0027) CMPRs. The same trend was observed during week 2. This trend was consistent with typical biocidal activity of new copper pipes [19]. Iron also removes DO from the water through redox reactions, leaving less for use by cells for respiration and growth [44, 46]. This trend occurred until week 9, when no difference was observed between PVC and PVC-copper total cell counts; although they still had higher total cell counts than PVC-iron CMPRs ($n_{each}$ = 18, p<0.0001 for both). This indicates that the copper pipes had aged to a point where the biocidal activity had decreased [18], potentially due to the formation of a more insoluble scale on the pipe wall. PVC-iron pipes; however, were found to still have ~1/3 of the DO of the other two conditions, potentially leading to lower cell growth due to nutrient limitations. There were no differences found among pipe types when comparing biofilm or bulk water qPCR measurements of 16S rRNA genes, which is likely due to inherent differences in the targets of the methods (DNA versus particles) [23].

*16S rRNA gene amplicon sequencing.* showed limited variability in microbial community composition among replicate CMPRs according to NMDS analysis. Blank samples were low in DNA content, as would be expected, with only one out of four blanks (one of the filter blanks) remaining in the analysis pool after rarefying to 10,177 reads across samples. This blank was taxonomically similar to samples ($R^2$ < 0.05) and was excluded under the assumption that it was cross-contaminated (S3 and S4 Figs). Statistically, there was a difference in beta diversity between the influent water samples and bulk water samples from all CMPRs ($P_{adonis}$ = 0.001 for both weighted and unweighted analysis); however, the magnitude of the difference was very small ($R^2$ = 0.093; $R^2$ = 0.068, unweighted and weighted, respectively). Differences were also noted between the bulk water and biofilm communities, although the magnitude was low ($P_{adonis}$ = 0.001, $R^2$ = 0.098, $P_{betadis}$ = 0.011). This difference is consistent with a previous study comparing bulk water and biofilm taxonomic composition in a continuous flow premise-plumbing rig comparing different pipe materials [17].

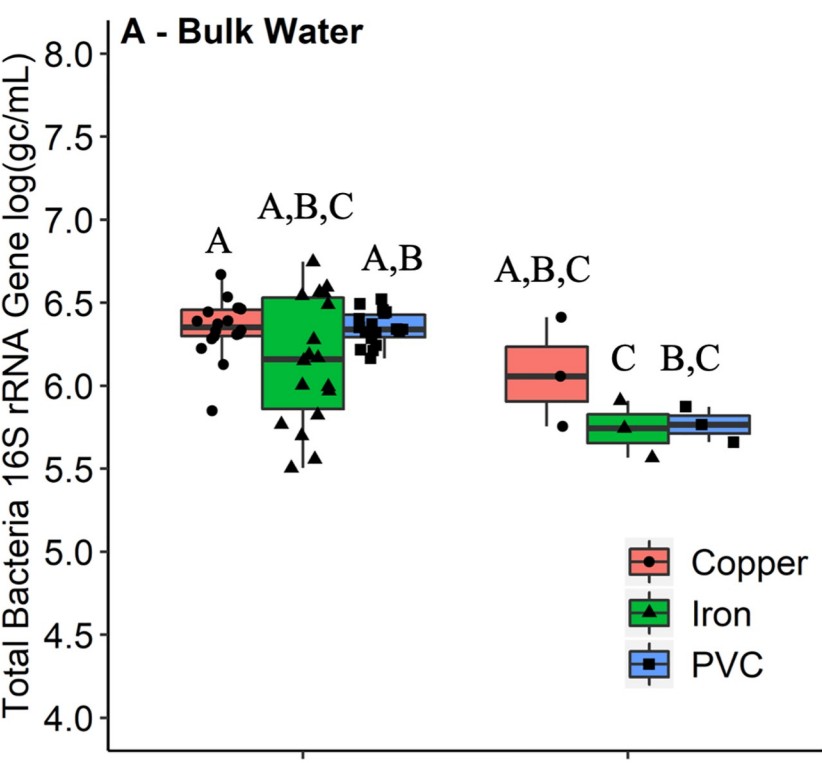

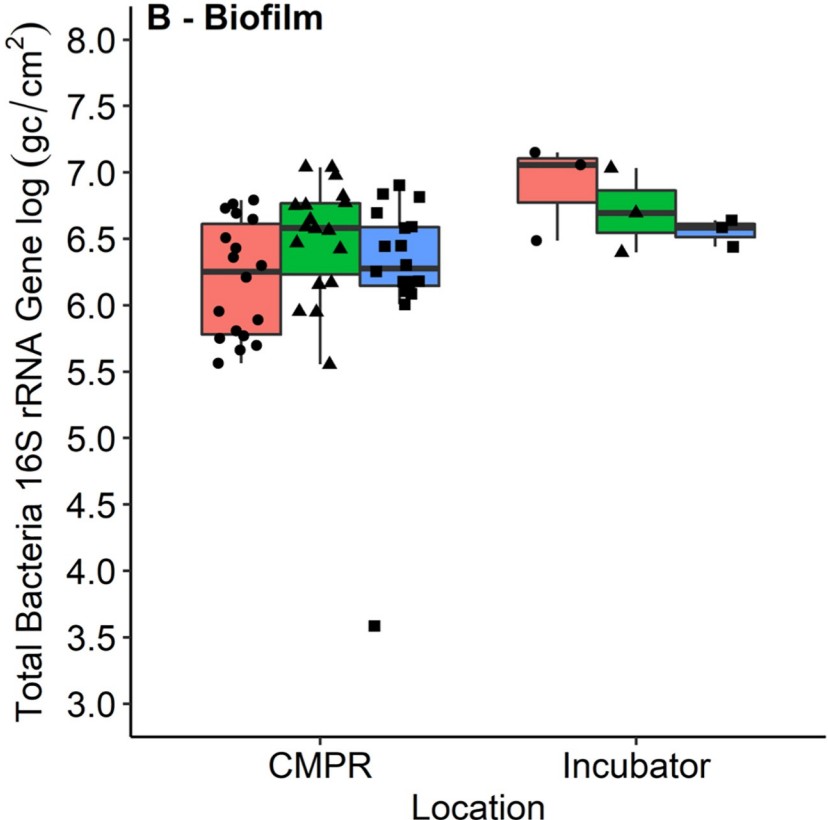

**Fig 3. Quantitative PCR comparison between CMPRs and stagnant pipes.** Comparison of (A) bulk water 16S rRNA DNA, and (B) biofilm 16S rRNA DNA, between CMPRs (n = 18) and stagnant incubator room pipes (n = 3) after 10 weeks of aging. Letters indicate groupings based on Dunn's test results (p < 0.05). Statistical groupings are independent for each panel.

Pipe type strongly influenced the composition of the microbial community that established in both the bulk water and biofilm, as has been observed in drinking water distribution systems [57], as well as other premise plumbing models such as pilot-scale rigs [58], simulated water heaters [59], and CDC biofilm reactors [60]. For CMPR bulk water samples, pipe type was found to be a major driver of microbial community composition ($P_{adonis}$ = 0.001, $R^2$ = 0.355, $P_{betadis}$ = 0.257) (S5A Fig). The microbial communities' shift in response to pipe material may explain why total cell count variation decreased in PVC-copper and PVC-iron pipes between weeks 2 and 9, while the putative drivers of total cell count variation in these pipes remained more consistent. Very few taxa were found to be significantly enriched in metal CMPR bulk waters, with eight unique ASVs enriched in PVC bulk water. Two ASVs of the order Sphingobacteriales and one of the order Solibacterales, which have been found to be elevated in sediment samples [61], were enriched in PVC-iron. One ASV of the order Lactobacillales was enriched in PVC-copper CMPRs. The tendency for ASVs to be enriched more in the PVC bulk water may have to do with either nutrients leaching from PVC, the release of copper in PVC-copper CMPRs, or the reduced DO in PVC-iron CMPRs. The same relationship with pipe type was found for CMPR biofilm samples ($P_{adonis}$ = 0.001, $R^2$ = 0.309, $P_{betadis}$ = 0.001); but there was more heterogeneity of variance in the measurements (S5B Fig). Conversely, 30

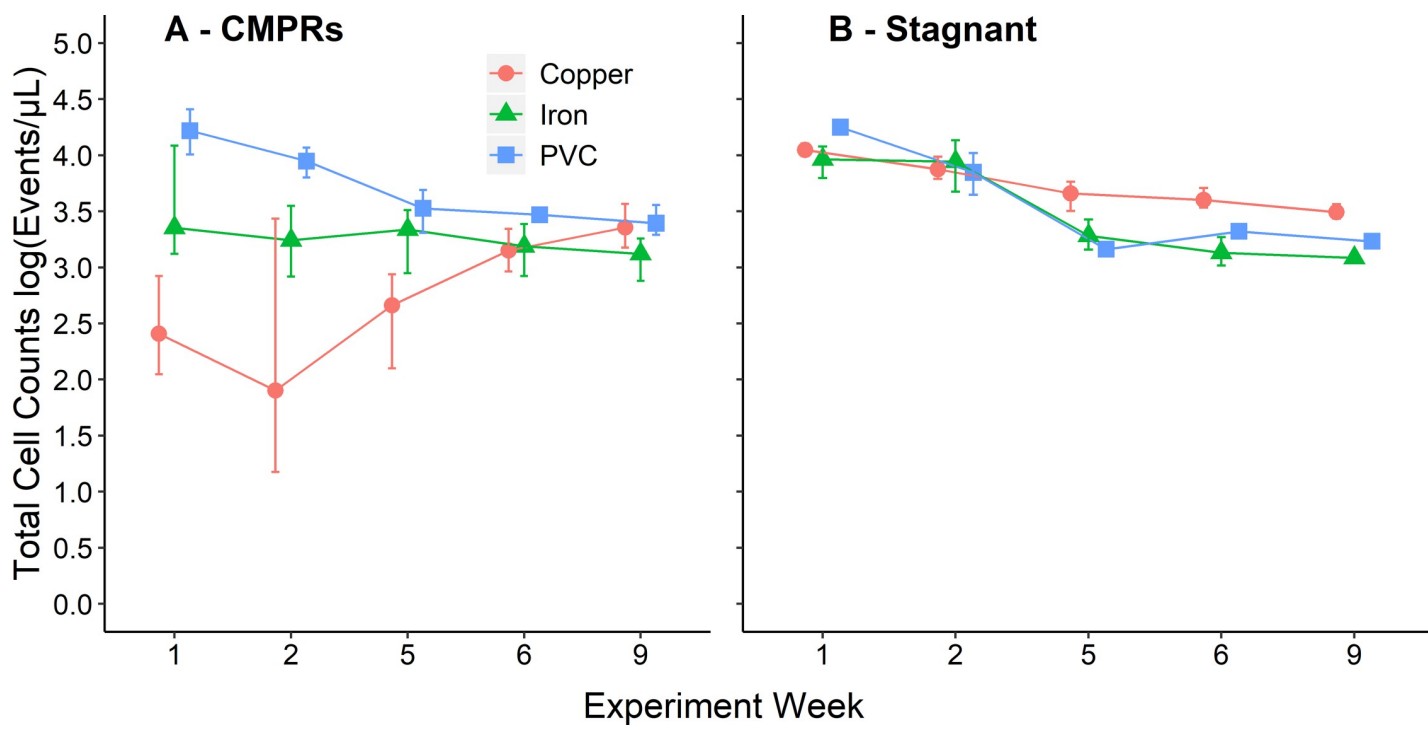

**Fig 4. Total cell counts in CMPRs and stagnant pipes as a function of aging.** Total cell counts via flow cytometry as a function of experimental week during the course of pipe aging for both (A) CMPRs (n = 18 per pipe type) and (B) stagnant pipes (n = 3 per pipe type) maintained in a constant temperature room at a 60° angle. For week 6, random subsets of n = 3 for PVC-copper and PVC CMPRs and n = 4 for PVC-iron CMPRs were analyzed. Week 1 and week 2 represent a single water change occurring for a given week. Following week 2, water changes increased to twice weekly. The observed change in cell counts over the course of the experiment is likely due to aging of new pipes. Error bars represent the range of measurements made on replicate pipes to show the full variability of a given group per week.

unique ASVs were found to be enriched in the biofilm of PVC-iron CMPRs, while none were enriched in PVC-copper CMPRs and only one ASV of the order Sphingomonadales was enriched in the PVC CMPR's biofilm. This may be due to the adherence of bacteria to the particulate iron on the pipe wall of PVC-iron CMPRs creating a niche environment for bacteria [62]. There was also no apparent difference in how PVC and PVC-copper biofilm samples clustered according to NMDS analysis, with the PVC-iron biofilm samples displaying the greatest variance (S5B Fig). This heterogeneity may be due to the age of the biofilm being only 10 weeks, while it is thought that at least one year is required to develop a truly mature biofilm on a copper pipe surface [63–65]. The greater variability in PVC-iron CMPRs may also be due to the influence of the sediments observed in the effluent of those pipes on the microbial ecology [62].

Alpha diversity based on the Shannon index was found to be lowest in the bulk water of PVC-copper CMPRs (1.60±0.37) and highest in PVC CMPRs (2.44±0.48), with PVC-iron CMPRs measuring in between (2.03±0.32) ($n_{each}$ = 18, PVC-copper vs PVC-iron, p = 0.0018; PVC-copper vs PVC, p<0.0001; PVC-iron vs PVC, p = 0.0018). This is likely due to a less diverse set of microbes being able to survive in the presence of the copper ions, as well as possible nutrients being released by the PVC pipe. The decrease in DO in the PVC-iron pipes may also contribute to lower diversity of microbes being able to thrive, leading to some taxa dominating over others. This trend was not observed in the biofilm, with no difference between PVC-copper and PVC pipes and the Shannon index being slightly higher in the PVC-iron pipes than in PVC-copper and PVC ($n_{each}$ = 18, p = 0.0253, p = 0.0027, respectively). The Shannon index was 1.43 ± 0.60 in PVC, 2.23 ± 0.55 in PVC-iron, and 1.73 ± 0.44 in PVC-copper CMPRs. This may be due to all samples being swabbed from the PVC portion of the biofilm, or the relatively low age of the biofilm itself.

## CMPRs versus stagnant pipes

**Physicochemical differences.** Following 9 weeks of aging, DO was consistently lower in the CMPRs than stagnant pipes, with PVC-iron pipes having the most pronounced difference (Fig 2A). The DO in PVC-iron CMPRs was 33% that of the PVC-iron stagnant pipes (1.96 mg/L vs 6.03 mg/L, respectively) ($n_{CMPR}$ = 6, $n_{stagnant}$ = 3, p = 0.020), yet there was no measurable difference between total iron levels in PVC-iron CMPRs (10.9 mg/L) compared to stagnant (10.8 mg/L) pipes. The DO in the stagnant PVC-iron pipes was comparable to that of the other two stagnant pipe conditions, indicating that the loss of DO to iron oxidation was reduced by stagnation. This suggests that the convective mixing current subjected the mild steel pipe section to more corrosive conditions. Convective mixing likely increased the interactions of the bulk water with suspended solids and biofilm, increasing mass transfer and reactions with corrosion byproducts that consume DO. Since water in the stagnant pipes does not actively mix, there are mass transfer limitations to biofilm and corrosion reactions that consume DO. The effect of lower DO in the CMPRs is desired in this case, as DO depletion is a common phenomenon in "real-world" water systems where galvanized iron pipes have corroded following the loss of the protective zinc layer [46].

There was no difference in TOC levels between CMPRs and corresponding stagnant pipes (Fig 2B). Following 9 weeks of aging, the pH was slightly higher on average in CMPRs than in stagnant pipes (7.34 vs. 7.06, $n_{CMPR}$ = 18, $n_{stagnant}$ = 3, p<0.0001) (Fig 2C). In terms of temperature, CMPRs inherently experienced a gradient between water bath (45–48˚C) and ambient (19–20˚C) temperatures, with an average of the mixture measured at 39.6˚C. The average measured temperature for the stagnant pipes was 35.2˚C, within the target optimal growth range for OPs (Fig 2D).

**Differences in microbial numbers.** In the stagnant pipes, total cell counts exhibited a different pattern from that of the CMPRs and there was no difference among the pipe materials, except PVC-copper had slightly higher total cell counts than in PVC-iron stagnant pipes starting at week 6 ($n_{each}$ = 3, p = 0.034) (Fig 4). This was the opposite of the trend displayed by the CMPRs, where PVC-copper pipe total cell counts were initially highly variable and much lower relative to PVC or PVC-iron pipes ($n_{each}$ = 18, p<0.0001, p = 0.0027, respectively). Interestingly, at week 2, although cell counts were ~2.5 logs lower in PVC-copper CMPRs than in PVC-copper stagnant pipes, there was no difference in concentration of total copper (0.66 mg/L vs. 0.46 mg/L) or soluble copper (0.23 mg/L vs. 0.33 mg/L). This was presumably due to convective mixing enhancing the interaction between cells and copper.

Differences in total cell counts between stagnant and CMPR PVC-iron conditions are likely explained by the differences in DO available to the bacteria (2.17 mg/L versus 6.03 mg/L at week 2, respectively). By week 9 the total cell counts were approximately equal in PVC-iron CMPRs and stagnant pipes, perhaps due to the water changes being increased to twice weekly (Fig 4).

PVC pipes had higher total cell counts in CMPRs than in corresponding stagnant pipes from week 5 onwards (Fig 4). This could be the result of recirculation of nutrients that is enhanced by convective mixing. Total bacterial 16S rRNA gene copy numbers were also higher in PVC CMPRs than in the stagnant PVC pipes ((6.36 log[gc/mL] vs. 5.78 log[gc/mL], respectively) ($n_{CMPR}$ = 18, $n_{stagnant}$ = 3, p = 0.007) during week 10 (Fig 3A). There were no differences in total cell counts for either PVC-copper or PVC-iron pipes in CMPRs relative to stagnant pipes at week 9, indicating diminishing pipe aging effects in the metal pipe CMPRs with time.

There was 4 times the density of bacterial 16S rRNA gene copies in stagnant PVC-copper pipe biofilm than in PVC-copper CMPR biofilm ($n_{CMPR}$ = 18, $n_{stagnant}$ = 3, p = 0.035), potentially due to less delivery of copper to the biofilm under stagnant conditions (Fig 3B). This difference was not found when comparing PVC or PVC-iron CMPRs and stagnant pipes.

The trend of higher total cell counts in PVC pipes, along with the trends observed in the metal pipes, is indicative of the impact that convective mixing has on the bulk water reactions in the interior of the pipe. Without internal convective mixing, the redox reactions and biocidal activity of the metals appear to have little to no effect on bacterial growth rates within the pipes of this particular water chemistry, although redox reactions may still have substantial effects under other circumstances. It also appears that more nutrients are also made available in the CMPRs than in the stagnant pipes. Overall, the results demonstrate that convective mixing can play an important role in bacteria-pipe-water interactions.

**Comparison of microbial communities in stagnant and CMPR Pipes.** There was an overarching difference in the microbial community composition of the bulk water of the CMPRs versus stagnant pipes based on the Bray-Curtis dissimilarity matrix ($P_{adonis}$ = 0.001, $R^2$ = 0.222, $P_{betadis}$ = 0.785), as well as the biofilm between the CMPRs and the stagnant pipes ($P_{adonis}$ = 0.001, $R^2$ = 0.119, $P_{betadis}$ = 0.827) (Fig 5). There was also a difference in the microbial community composition when considering the unweighted Jaccard similarity matrix in the bulk water ($P_{adonis}$ = 0.001, $R^2$ = 0.148, $P_{betadis}$ = 0.966) and biofilm ($P_{adonis}$ = 0.001, $R^2$ = 0.088, $P_{betadis}$ = 0.695). Comparing PVC-copper pipes, there were 84 unique ASVs enriched in the bulk water of stagnant PVC-copper pipes compared to only one enriched in PVC-copper CMPRs. This suggests stronger biocidal effects in copper CMPRs than stagnant pipes. For PVC-iron pipes, 5 ASVs were enriched in stagnant pipes and 21 in CMPRs. For PVC pipes, 12 ASVs were enriched in stagnant pipes and 15 in CMPRs. Overall, there were 58 enriched ASVs in the biofilm of stagnant pipes compared to CMPRs, whereas only 9 were enriched in the biofilm of CMPRs. Since biofilm was collected from the bottom of the PVC portion of each pipe, it is possible that the bacteria were more likely to settle in the stagnant pipes than the CMPRs, allowing more adherence to the pipe wall at that location.

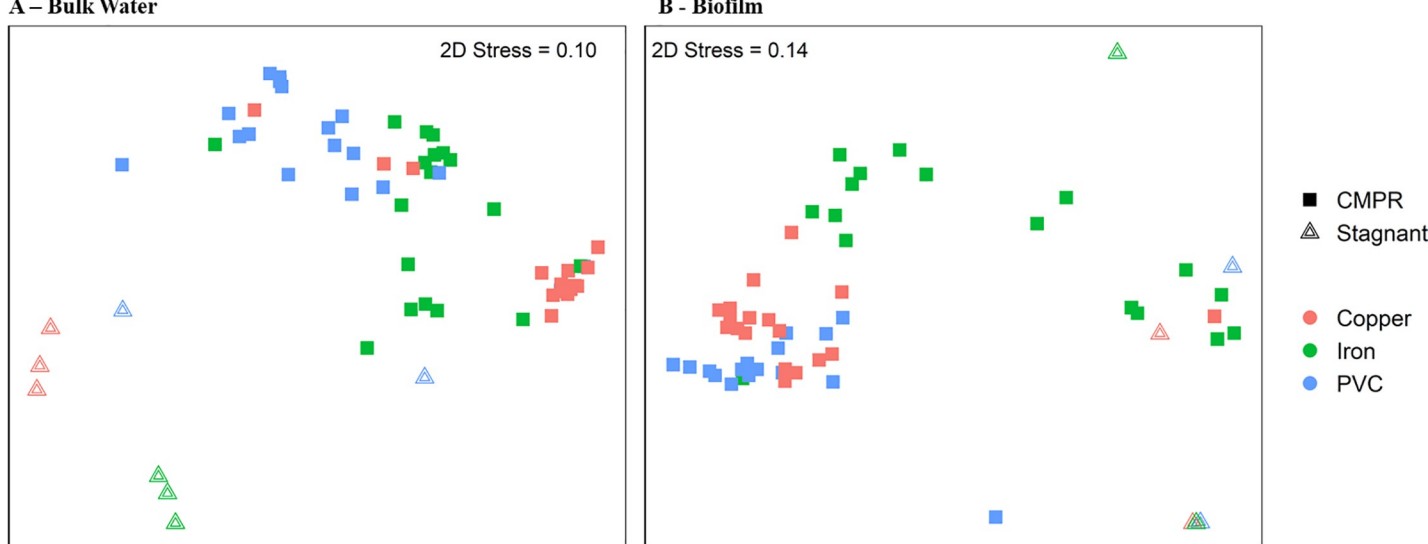

**Fig 5. Dissimilarity in microbiome compositions of samples between CMPRs and stagnant pipes.** Nonmetric multidimensional scaling (NMDS) plot, generated from Bray-Curtis dissimilarity matrix using the phyloseq package in R for 16S rRNA gene amplicon sequences, comparing CMPR and stagnant pipe (A) bulk water and (B) biofilm taxonomic microbial community composition at the end of week 10. Bulk water and biofilm NMDS plots were generated independently. There is a difference in microbial community when comparing both the bulk water $P_{adonis}$ = 0.001, R2 = 0.222, $P_{betadis}$ = 0.785) and biofilm ($P_{adonis}$ = 0.001, R2 = 0.119, $P_{betadis}$ = 0.827) of CMPRs vs stagnant pipes.

Alpha diversity in the stagnant pipes was not found to be affected by pipe type for either the bulk water or the biofilm. Overall, the Shannon index was 2.92 ± 0.30 for bulk water samples and 2.79 ± 1.10 for the biofilm samples. This is in contrast to the trend measured in CMPRs noted above, where alpha diversity was lowest in PVC-copper pipes, indicating again that convective mixing enhanced the effects of metal pipes.

### CMPRs versus other premise plumbing simulation methods

CMPRs were designed to deliver nutrients (e.g., oxygen, organic carbon) to biofilms in the low-nutrient environment of drinking water while maintaining ideal temperatures for OP growth throughout the pipe volume with controlled replication. In this study, the CMPRs achieved a relevant premise plumbing water chemistry conditions while using more representative pipe materials than the simulated glass water heaters (SGWHs) or CDC biofilm reactors, and allowing for more precise control of conditions than pilot-scale premise plumbing systems (Table 2).

While costs of any system will vary with time and location, the test apparatus described herein, including the housing unit, cost only ~$1,200 to accommodate 54 parallel operated CMPRs (S3 Table). Pilot-scale premise plumbing rigs constructed in prior studies were estimated to cost approximately ~$4,000 and accommodate fewer replicates (e.g., 36 pipes) and thus had more inherent variability because each distal line is not a perfect replicate [10]. CMPRs are also much less costly than the CDC biofilm reactor, with a cost of $2,480 USD per reactor with stir plate and temperature controls, not including a peristaltic pump needed for each reactor (Biosurface Technologies, Bozeman, MT). This makes testing multiple pipe conditions in parallel expensive and often cost prohibitive, as an individual reactor with full controls will be needed for each pipe type tested, while also introducing a greater chance of mechanical failure of at least one component. Moreover, when a failure of a component inevitably occurs in a CMPR, true replication is maintained because it will affect every pipe equally.

**Table 2. Comparison of the CMPR to alternative reactors for simulating premise plumbing based upon available pricing and other information for required equipment components and personal knowledge and experience of the authors.** Y = Yes, N = No, D = difficult/expensive.

| | CDC Biofilm Reactor | Simulated Glass Water Heaters (SGWHs) | Pilot-Scale Premise Plumbing Pipe Rigs | Convective Mixing Pipe Reactors (CMPRs) |
|---|---|---|---|---|
| Precise control of influent water conditions? | D | Y | N | Y |
| Ease of operation under premise plumbing conditions (simulates extremes in water quality/flow changes)? | N | Y | Y | Y |
| Realistic flow conditions? | N | N | Y | N |
| Standardized protocol and methods? | Y | Y | N | Y |
| Multiple conditions simultaneously? | D | Y | D | Y |
| Multiple pipe materials simultaneously? | D | Y | D | Y |
| Replicate reactors practical? | N | Y | N | Y |
| Ease of replication? | Y | Y | N | Y |
| Economical to build? | N | Y | N | Y |
| Off the shelf components only? | N | Y | Y | Y |
| Mostly representative materials? | N | N | Y | Y |
| High pipe surface area/volume ratio? | N | N | Y | Y |
| Low manual labor? | D | N | N | N |
| Small footprint? | D | Y | N | Y |
| Low cost to operate/maintain? | N | Y | N | Y |
| Amenable to biosafety protections? | D | Y | N | Y |
| Cost, USD (S3 Table) | $2,480 /reactor | $3 /bottle | ~$4,000 /rig | ~$1,200 /rig |
| Representative Cost, USD per replicate reactor | $2,480 | $3 | $450* | $22 |

*Assuming each distal line from recirculation rig is a separate reactor, otherwise $4,000

Although SGWHs are the most economical option (~$3 USD/reactor), the vast majority of the available surface is glass, unless steps are taken to equip with extended pipe surface area. SGWHs are also largely stagnant, unless placed on a shaker table, but the extent to which RPMs correspond to real-world premise plumbing conditions has not been determined.

The ultimate basis of comparison, however, should be the ability of these systems to replicate real-world premise plumbing conditions. Here, we have shown that the CMPRs replicate phenomena observed in some important real-world settings. Unfortunately, it is not possible to compare our water quality findings to those of other full-scale systems because every water has a unique chemistry and every sampling location in a building is unique in terms of water use patterns, temperature and disinfectant time profiles. Such a comparison would be useful but it would be a major undertaking in terms of personnel and resource requirements, but would be highly valuable because it is important to obtain a better understanding of the real-world strengths and limitations of each laboratory simulation method. A particular strength of CMPRs is an ability to be modified to accommodate a wide range of replicates, pipe conditions, temperature regimes, and adjustments to influent water chemistry. We judge that CMPRs are a useful, novel and cost-effective test method that can help to better understand a wide variety of microbiological and physicochemical premise plumbing research issues.

## Conclusions

Herein we created and tested a novel reactor design that takes advantage of a natural convective mixing phenomenon to facilitate replicated testing of realistic premise plumbing conditions for purposes such as evaluating conditions that support pathogen growth or disinfection. Several key aspects of the CMPRs were validated:

- CMPRs can maintain temperatures ideal for the growth of OPs.

- Little variability in physical and biological parameters were observed across replicate CMPRs beyond that which is inherent to new pipe materials, allowing for relatively homogenous testing conditions.

- Without convective mixing, there was little to no effect of redox reactions or biocidal activity normally expected when there are interactions with metal pipes, demonstrating that such mixing can be an important factor for bacterial growth and pipe interactions when present.

- CMPRs are a cost-effective system that can simulate a substantial range of premise plumbing conditions.

- CMPRs were able to induce and isolate certain effects of key test variables, e.g., pipe material effect on water chemistry and microbiology, that have been observed as trends in other premise plumbing simulations and full-scale buildings. These include chemical phenomena such as the observed differences in pH and DO, the variability and gradual reduction in copper and iron release, as well as biological differences such as total cells counts and microbial community composition in the bulk water and biofilm. This enables future testing to explore impacts of a wider range of chemistries and experimental conditions while rigorously testing for statistical confidence between true replicate reactors.

Overall, the CMPR provides a simple, replicable reactor capable of simulating some important naturally occurring phenomenon representative of premise plumbing.

## Supporting information

**S1 Fig. Temperature profile of clear PVC pipe filled with RO water after allowed to equilibrate with the room and incubator temperatures to undergo convective mixing at different angles.** Incubator consisted of a box using a light bulb as a heat source. Temperatures were measured using an infrared thermometer temperature gun at various points along the pipe. (DOCX)

**S2 Fig. Comparison of total cell counts, DO, and total iron in CMPR effluent in weeks 2 and 9.** No significant correlation was found between total cell counts and either DO or total iron (Spearman rank correlation). (DOCX)

**S3 Fig. Phylum level taxonomic microbial community profiles of each CMPR and each stagnant pipe organized by pipe type and location.** Naming convention: the first letter is a nominal indicator representing a set of 9 pipes representing a row of pipes in the CMPR housing unit, the second letter indicates pipe type (C = copper, F = PVC-iron, P = PVC), and the number represents the replicate number within the set of 9 pipes. All samples were collected during Week 10 with the exception of influent and seeded influent samples (Week 0). (DOCX)

**S4 Fig. Nonmetric multidimensional scaling (NMDS) plot, generated from Bray-Curtis dissimilarity matrix using the phyloseq package in R for 16S rRNA gene amplicon sequences rarefied to 1139 randomly selected reads to maintain all blanks and samples to compare taxonomic microbial community composition between blanks and samples.** A single blank (filter blank) clusters near the majority of samples. This is assumed to be the result of cross-contamination during the preparation process given that all other blanks cluster separately. The microbial community compositions of blanks were found to be different than that of both bulk water and biofilm samples ($P_{adonis} < 0.01$ for both unweighted and weighted

metrics).
(DOCX)

**S5 Fig.** Nonmetric multidimensional scaling (NMDS) plot, generated from Bray-Curtis dissimilarity matrix using the phyloseq package in R for 16S rRNA gene amplicon sequences, comparing CMPR (A) bulk water and (B) biofilm taxonomic microbial community composition. Bulk water and biofilm NMDS plots were generated independently. Pipe type was an important factor influencing the microbial community in both the bulk water $P_{adonis} = 0.001$, $R^2 = 0.355$, $P_{betadis} = 0.257$) and biofilm ($P_{adonis} = 0.001$, $R^2 = 0.309$, $P_{betadis} = 0.001$).
(DOCX)

**S1 Table. Test data determining the maximum mixing velocity based on pipe angle using neutrally buoyant rhodamine dye.**
(DOCX)

**S2 Table. Physicochemical characteristics of influent water.**
(DOCX)

**S3 Table. Cost estimates for each type of premise plumbing simulation method.**
(DOCX)

**S1 File. Manuscript data.** Data used for analysis in this manuscript.
(XLSX)

**S2 File.**
(DOCX)

# Acknowledgments

The authors acknowledge Advanced Research Computing at Virginia Tech for providing computational resources and technical support that have contributed to the results reported within this paper. The authors would like to thank Sophia Lee for their assistance with maintenance of the CMPRs and data collection.

# Author Contributions

**Conceptualization:** M. Storme Spencer, William J. Rhoads, Marc A. Edwards.

**Data curation:** M. Storme Spencer.

**Formal analysis:** M. Storme Spencer, Abraham C. Cullom.

**Investigation:** M. Storme Spencer, Abraham C. Cullom, William J. Rhoads.

**Methodology:** M. Storme Spencer, William J. Rhoads, Amy Pruden, Marc A. Edwards.

**Resources:** Amy Pruden, Marc A. Edwards.

**Supervision:** William J. Rhoads, Amy Pruden, Marc A. Edwards.

**Visualization:** M. Storme Spencer.

**Writing – original draft:** M. Storme Spencer.

**Writing – review & editing:** Abraham C. Cullom, William J. Rhoads, Amy Pruden, Marc A. Edwards.

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
