## [Decision Letter · Decision Letter 0]

12 Jun 2020

PONE-D-20-13252

Replicable simulation of distal hot water premise plumbing using convectively-mixed pipe reactors

PLOS ONE

Dear Dr. Edwards,

Thank you for submitting your manuscript to PLOS ONE. After careful consideration, we feel that it has merit but does not fully meet PLOS ONE’s publication criteria as it currently stands. Therefore, we invite you to submit a revised version of the manuscript that addresses the points raised during the review process.

It's an interesting study to use the new reactor to study plumbing systems. As mentioned by the reviewer, more experimental details can be provided and experimental conditions can be better justified. Showing how the new CMPRs represent or be similar to real plumbing systems would greatly strengthen the paper. Please address other detailed comments from the reviewer as well.

We look forward to receiving your revised manuscript.

Kind regards,

Zhi Zhou, Ph.D.

Academic Editor

PLOS ONE

Journal Requirements:

Additional Editor Comments (if provided):

Reviewers' comments:

Reviewer's Responses to Questions

**Comments to the Author**

1. Is the manuscript technically sound, and do the data support the conclusions?

Reviewer #1: Partly

2. Has the statistical analysis been performed appropriately and rigorously? 

Reviewer #1: Yes

3. Have the authors made all data underlying the findings in their manuscript fully available?

Reviewer #1: Yes

4. Is the manuscript presented in an intelligible fashion and written in standard English?

Reviewer #1: Yes

5. Review Comments to the Author

Reviewer #1: This paper describes a new approach to simulating domestic plumbing systems that are influenced by convective mixing. The introduction was interesting and informative. Some parts of the methods section were well done, but other parts were lacking. Specifically, some important experimental details were included in the results section rather than in the methods and some of the reasons for some of the experimental design decisions were not explained (e.g. Why were some parameters measured weekly and others measured only in week 2 and week 9? Why was the water changeover schedule changed from once a week to twice a week partway through the experiment?) The results section is comprehensive but somewhat disorganized. For the most part, the results supported the conclusions at the end of the paper, but it could be better organized and written to emphasize how each set of experiments informed the final conclusions.

General comments:

My main problem with this work is that the authors failed to put their results into context or to prove that their CMPRs were representative of full scale domestic plumbing systems, or comparable/superior to alternative pilot systems (e.g. pilot scale pipe rigs, model showers), or existing bench-scale reactors (e.g. CDC reactors). This made it impossible to evaluate whether their CMPRs were representative of real world conditions and/or more appropriate than alternative pilot-scale and bench-scale options. In theory, the best way to explore this would be by conducting parallel studies with the different simulation options and/or in real domestic hot water systems. This may not be feasible at this point, but the authors are part of a large, well established laboratory and, based on other papers published out of this group, should have access to data and expertise that could substantially improve this aspect of the paper.

The section devoted to comparing the CMPRs with other simulation options focused mostly on cost rather than water quality and biological impacts. I think it would be much stronger if they focused more on the latter, as described in the specific comments later in this review.

There seems to have been a large amount of variability between the individual CMPRs within each condition (i.e. each pipe type) for some parameters (Table 1). This contradicts one of the conclusions at the end of their paper. The coefficient of variation for temperature was low for all conditions, so presumably the variability was not related to temperature management. I don’t feel like the authors convincingly explained why replicate new pipes exposed to identical conditions would have such different results.

Finally, the writing was often convoluted with small but frequent grammatical errors. A careful review of the text should be sufficient to remedy this. The flow of the paper could be tightened by ensuring that the results section is organized such that the different sections, including the final section comparing the CMPRs to other simulation options, clearly support the final conclusions of the study.

Specific comments:

Abstract Is it standard to include n, p, and R2 values in abstracts in PLOS ONE? It seemed odd to me when I was reading through the paper, but if it is normal for this journal there is no need to change it.

Line 44 Emanating is a strange word choice here. Perhaps originating would be more appropriate.

Lines 55-56 Impeded from replication is an awkward way to say that it is difficult to include large numbers of replicates. I suggest that you reword this.

Lines 59-60 Would biological safety standards be difficult to maintain because the pilot equipment is likely to be too large for a standard laboratory? If a pilot system was small enough to fit in a laboratory, would it still be difficult to maintain biological safety standards?

Lines 82-85 This sentence is convoluted and too long. I suggest that you break it into two or three

smaller and more focused sentences.

Lines 111-112 The copper + PVC CMPRs referred to as copper pipes throughout the text but the iron + PVC pipes are referred to PVC-iron. I suggest that you refer to the copper + PVC pipes as PVC-copper throughout the paper to emphasize that, like the iron pipes, the copper pipes were connected to a PVC portion.

Line 115 Aren’t the different properties of the materials the point of the study?

Lines 115-116 Why did you include so many replicates?

Lines 119-120 I don’t think that once or twice weekly water changes are likely to replicate water usage patterns in real domestic plumbing systems where water is used many times per day. Also, why did you change from once a week to twice a week partway through your experiment?

Line 125 Could you include some basic information about the “base” (influent?) water that you used in your experiments (TOC, iron, etc.) in a table in the supplementary information? Also, you refer to this water as base water in some parts of the paper and influent water elsewhere. Choose one term and stick to it.

Line 128 No need to put the word seeded in quotes here or elsewhere.

Line 129 Is the bacterial community in GAC backwash water likely to be representative of that found in a domestic plumbing system? If yes, could you explain why?

Lines 143-144 Why were these parameters only measured in week 2 and week 9? Why not measure them more frequently?

Line 169 By circular motion do you mean that you swabbed the entire inner circumference of the pipe or that you took a swab from one part of the pipe (e.g. the bottom surface)?

Line 194 Please describe the different blanks mentioned here.

Lines 219-220 You could remove this sentence.

Line 231 What are the expected pH trends for each type of pipe? Please describe briefly and provide citation(s). If these expected trends were observed in full scale domestic water heating systems it might be worth mentioning this in the final section of the results to support the idea that your CMPRs are comparable to real world systems.

Lines 280 Why would individual iron pipes age at different rates if they are all new and all subjected to the same conditions?

Lines 283-286 Break this sentence into two sentences, one about copper and one about iron.

Lines 284-285 Why would different individual copper pipes release different amounts of cupric ions if they were all new and all subjected to the same temperature conditions and water changeout regimes? This potentially points to a serious weakness in your approach or the execution of the study.

Lines 312-314 Add a citation to support this hypothesis.

Lines 314-315 Have other studies found that iron pipes have lower DO than copper or PVC pipes?

Lines 322-324 This should also be mentioned in the methods section.

Lines 324-325 The fact that new pipes were used does not indicate that pipe aging occurred, but the fact that you started with new pipes and your cell counts changed over time does suggest that pipe aging occurred. I suggest that you reword this sentence to clarify.

Lines 325 Have other studies shown that pipe aging impacts cell counts? If yes, cite them here (you cited refs 37 and 38 to support a similar statement elsewhere in the paper). Depending on what kind of apparatus was used in these other studies, this could also be discussed in the final part of the results section.

Line 324 Why did you change your water changeover regime in week 2?

Line 330 Remove the word inherently.

Line 334 When you say “bulk water from the CMPRs”, do you mean that all of the data from all of the conditions (PVC, iron-PVC, copper) were grouped together and compared to the influent water? If yes, I would mention this in the text.

Lines 341-350 Has this relationship between microbial community and pipe material been observed in real world domestic hot water systems or in studies conducted with other lab scale apparatus? If yes, this would be an interesting thing to bring up in the final section of the results.

Line 344 Remove the word however.

Line 361 Did you actually observe sediments in your pipes?

Line 372 Why were all of the swabs taken from the PVC section of the pipes?

Lines 382-385 I don’t think that your DO findings, on their own, support all of this. I suggest that you tone down your language and bit and break this sentence into two sentences. The first sentence could begin with “this suggests that” rather than “thus” to emphasize that you are, to some degree at least, speculating about the causes underlying your DO results.

Lines 386-389 This is an interesting point. Perhaps you could move it to the final section of the results to better support your assertions that the CMPRs can simulate real world conditions.

Line 418 Here and elsewhere in the paper you should write times, rather than x.

Lines 459-482 This section is, in my opinion, the heart of the paper, but it contains a number of unsupported or poorly supported claims by the authors and as a result it is ultimately unconvincing. For example, reference 10 was used to support the statement that pilot-scale plumbing rigs had higher inherent variability than your CMPRs, but when I quickly reviewed that paper, I saw no mention of high variability.

In general, this section would be more effective if you focused on comparing your actual water quality findings (e.g. DO, cell counts, etc.) to the results of studies that examined full scale domestic hot water systems or that used CDC reactors, pilot pipe rigs, etc.. This would show that your CMPRs were comparable or superior to existing methods. There is some of this in various parts of the results section (lines 338-340, lines 386-389, etc.) and this content could be moved to this final section to

better support your claims.

The cost information is interesting but ultimately costs will differ from one jurisdiction to another and over time, so I wouldn’t make it the main focus here.

Also, assuming that stagnant pipe reactors like those used in this study are widely used to simulate domestic hot water systems, they should be included in Table 2 and discussed in this section.

Line 460 You didn’t measure nitrogen or discuss it in the rest of the paper, so I suggest that you omit it from this list.

Line 461 This is the first time that your base / influent water is described as oligotrophic. This term may not be familiar to your readers, so I think it would be more effective to briefly describe the important characteristics of your water matrix (low TOC, moderate pH, dechlorinated, etc.)

Line 476 Are mechanical failures truly unavoidable with the CDC reactors? Couldn’t the UV light, recirculation pump, heater, etc. in the CMPR set-up fail as well?

Line 489 Some of your results, especially from week 2, do not support this conclusion.

Figure 1 What does the checked box represent? Also, it might be nice to include a schematic showing the stagnant pipe apparatus for comparison.

Figure 2 I must admit to initially being totally confused by the letters on these plots because the way that you’ve indicated statistical differences is unfamiliar to me and, in my opinion, counterintuitive. I’m used to statistical differences being indicated by different letters, rather than by the same letters. So, for example, if condition 1 and condition 2 were statistically different, they would be labeled A and B, but if they were indistinguishable, they’d both be labeled A because they would be in the same group. This is in line with the outputs of all of the statistical programs that I’ve ever worked with. I think that the approach that I’ve described here is likely to be more familiar and intuitive to readers, so I suggest that you adopt it here.

Also, why do some plots only have capital letters and some only have lower case letters? Weren’t the same statistical tests run on all four datasets?

Figure 4 It would be interesting to see DO, cell counts, and iron plotted together (or on three faceted plots in a single figure) to emphasize how they influence one another.

Table 2 This table is interesting but the authors don’t support their assumptions with any citations and it isn’t always clear how they came to their conclusions about different factors. For example, why wouldn’t it be possible create a standardized protocol for a pilot-scale pipe rig? Why would the CMPRs cost less to operate than a CDC reactor, especially seeing as the CMPR apparatus includes a UV lamp and heater, both of which would consume electricity? Why would it be more difficult or expensive to ensure biosafety protections for CDC biofilm reactors? Also, the authors should specify that their costing is USD.

6. PLOS authors have the option to publish the peer review history of their article (what does this mean?). If published, this will include your full peer review and any attached files.

Reviewer #1: No

---

## [Author Response · Author response to Decision Letter 0]

27 Jul 2020

Our responses are also uploaded in a file with better formating to view each comment. We also produce it below.

We thank the editor and reviewer for their time and effort in evaluating our manuscript and offering suggestions for improvement. Attached are the original copy of the manuscript and a revised version using tracked changes. Responses to comments are provided below with the line numbers referring to the track changes version of the manuscript. Portions of quotations are bolded to emphasize particular changes to the editors and do not reflect bold formatting in the manuscript.

Reviewer 1:

General comments:

1-1 My main problem with this work is that the authors failed to put their results into context or to prove that their CMPRs were representative of full scale domestic plumbing systems, or comparable/superior to alternative pilot systems (e.g. pilot scale pipe rigs, model showers), or existing bench-scale reactors (e.g. CDC reactors). This made it impossible to evaluate whether their CMPRs were representative of real world conditions and/or more appropriate than alternative pilot-scale and bench-scale options. In theory, the best way to explore this would be by conducting parallel studies with the different simulation options and/or in real domestic hot water systems. This may not be feasible at this point, but the authors are part of a large, well established laboratory and, based on other papers published out of this group, should have access to data and expertise that could substantially improve this aspect of the paper.

Response: We agree with the reviewer that such testing would be ideal, but that it is not feasible to do in a meaningful way because every feed tap water is inherently different and the experiments with different designs would have to be done in parallel. This is a main message of the manuscript, that no existing system reasonably allows for such testing in parallel at the scale of the CMPR. We note that such testing has also not previously been reported for any of the alternative laboratory simulation highlighted in the manuscript. 

In response to this comment, we point out some of the obvious challenges that would have to be overcome for such testing in lines 754-760.

Additionally, we followed the advice of the reviewer to the extent possible with new text in lines 393-395, 439-440, 444-446, 500-501, 505-506, and 558-561, with the purpose of better contextualizing our work with comparisons to previous studies. Overall, changes have been made to emphasize that our water quality findings correspond reasonably well to those of other simulation methods or full-scale buildings, but that it would be valuable to have a large-scale study to try and compare strengths and weaknesses of every lab simulation method to portions of actual premise plumbing in buildings.

Context has also been provided for the inherent variability of iron and copper release, which are a key concern for this reviewer. Briefly, new metallic pipes are expected to be variable in metal release rates. See lines 439-440 and 444-446 and responses to Comments 1-23 and 1-25 below for additional details.

1-2 The section devoted to comparing the CMPRs with other simulation options focused mostly on cost rather than water quality and biological impacts. I think it would be much stronger if they focused more on the latter, as described in the specific comments later in this review.

Response: See response to Comment 1-1 above and Comment 1-41 below. The reality is that cost and scale are major limitations to realistic premise-plumbing simulations. This study proposes and evaluates a new design, while also providing context to existing alternatives. 

1-3 There seems to have been a large amount of variability between the individual CMPRs within each condition (i.e. each pipe type) for some parameters (Table 1). This contradicts one of the conclusions at the end of their paper. The coefficient of variation for temperature was low for all conditions, so presumably the variability was not related to temperature management. I don’t feel like the authors convincingly explained why replicate new pipes exposed to identical conditions would have such different results.

Response: This reviewer concern has been addressed in detail below in response to Comment 1-23 and Comment 1-25. Additionally, we discuss the expectation that there is inherent variability in premise plumbing studies in response to Comment 1-12. 

1-4 Finally, the writing was often convoluted with small but frequent grammatical errors. A careful review of the text should be sufficient to remedy this. The flow of the paper could be tightened by ensuring that the results section is organized such that the different sections, including the final section comparing the CMPRs to other simulation options, clearly support the final conclusions of the study.

Response: All co-authors have re-read and revised the manuscript to address grammatical errors and improve clarity. We have worked to better present the results, as suggested, and to ensure that the results support the conclusions, as seen in our responses to Comments 1-22, 1-23, 1-25, 1-27, 1-28, 1-29, 1-30, 1-34, 1-38, and 1-39. Additionally, we have added a new co-author, Abraham Cullom, who was previously an acknowledgement for assisting in testing the CMPRs, but now aided in independently re-analyzing the data and revising the manuscript with fresh perspective. Abraham is also applying the CMPR design in his current research towards answering some of the questions raised in the current study.

Specific comments:

1-5 Abstract Is it standard to include n, p, and R2 values in abstracts in PLOS ONE? It seemed odd to me when I was reading through the paper, but if it is normal for this journal there is no need to change it.

Response: We have evaluated other PLOS ONE papers and have determined that it is common to report statistical details in the abstract. Thus, we have maintained the statistical details in the abstract.

1-6 Line 44 Emanating is a strange word choice here. Perhaps originating would be more appropriate.

Response: ‘Emanating’ has been replaced with ‘originating’ in line 55.

1-7 Lines 55-56 Impeded from replication is an awkward way to say that it is difficult to include large numbers of replicates. I suggest that you reword this.

Response: The phase ‘are impeded from replication by size and cost’ has been replaced by ‘hinder replication through their large size and cost’ in lines 66-67.

1-8 Lines 59-60 Would biological safety standards be difficult to maintain because the pilot equipment is likely to be too large for a standard laboratory? If a pilot system was small enough to fit in a laboratory, would it still be difficult to maintain biological safety standards?

Response: A primary difficulty associated with pilot-scale systems of any size is that they require direct connection to the actual premise plumbing of a public building and associated institutional permissions and oversight for automated operation. By design, pilot-scale systems are continuous-flow and produce large volumes of water that must be assumed to be contaminated with pathogens and thus appropriately disinfected and disposed of. Typically, this requires inline disinfection and hard plumbing to the drain. Note that most of the relevant pathogens of interest to premise plumbing can be spread via inhalation of bioaerosols, creating a potential health risk during sampling. Because a pilot-scale reactor cannot fit within the confines of a biological safety level (BSL)-2-approved cabinet, sampling must be carried out in the open. To safely conduct sampling, a BSL-2 plan must be developed and approved by the institution, typically with the laboratory evacuated and the workers wearing N-95 masks. 

Lines 70-82 have been amended to clarify the biological safety concerns of pilot-scale systems. In particular, CMPRs can be sampled safely within the confines of a BSL-2- approved cabinet.

‘Pilot-scale studies examining OPs also typically require direct connection to premise plumbing of the study facility and cannot be sampled within the protection of a biological safety-level (BSL) 2 certified cabinet, elevating potential for exposure of workers to pathogen-containing aerosols during sampling and thus requiring appropriate institutional approvals.’

1-9 Lines 82-85 This sentence is convoluted and too long. I suggest that you break it into two or three smaller and more focused sentences.

Response: This sentence has been divided into two in lines 129-132:

‘The CMPR consists of off-the-shelf materials used in real-world plumbing systems. Capped pipe segments, with one end submerged in a hot water bath and the other contacting the cooler ambient air, simulate the premise plumbing riser from a hot water recirculation loop that connects to distal, stagnant outlets (Fig 1A).’

1-10 Lines 111-112 The copper + PVC CMPRs referred to as copper pipes throughout the text but the iron + PVC pipes are referred to PVC-iron. I suggest that you refer to the copper + PVC pipes as PVC-copper throughout the paper to emphasize that, like the iron pipes, the copper pipes were connected to a PVC portion.

Response: References to ‘copper CMPRs/pipes’ have been changed to ‘PVC-copper CMPRs/pipes’ throughout the text.

1-11 Line 115 Aren’t the different properties of the materials the point of the study?

Response: Differences in heat conduction properties of these materials were not the focus of this particular study, although examining how pipe material’s effects on internal mixing velocities affect a multitude of factors could be an important focus of a future project. This study attempted to isolate the properties of the three pipe materials that would influence bulk water chemical and biological parameters while maintaining the same internal mixing velocity. The text has been modified in lines 215-218 for clarification:

‘These three materials exhibited distinct heat conduction properties and would in turn create distinct convective mixing patterns and velocities if submerged in the water bath. Thus, for purposes of this experiment, PVC segments were submerged in the water bath to normalize convective mixing across the conditions.’

1-12 Lines 115-116 Why did you include so many replicates?

Response: A major limitation of all prior premise plumbing OPs research, including our own, is the insurmountable costs of obtaining true replication. That limitation reduces statistical confidence and increases the likelihood of drawing false conclusions or yielding statistically insignificant results. A strength of the CMPR approach is that replicates can be added are relatively low cost and, frankly, we were very curious as to how reproducible reactors and chemical and biological conditions could be with a large number of replicates using an approach that minimized most sources of variability to the extent possible. We suspect that if all prior work (including our own) had required a minimum n = 5 (which is cost prohibitive for most current premise plumbing simulations) that a high inherent variability of prior systems would become much more apparent. Since this is the first test of the CMPR concept, we wanted to also do the first test using a very large number of replicates to explore this issue, in an approach in which most contributors to variability were minimized. The variability of CMPRs was therefore a key focus of this study. 

This sentence has been prefaced with ‘to ascertain a high-resolution view of the variability of chemical and biological water quality parameters among CMPRs’ in lines 218-219. 

1-13 Lines 119-120 I don’t think that once or twice weekly water changes are likely to replicate water usage patterns in real domestic plumbing systems where water is used many times per day. 

Response: We agree that twice weekly water changes do not represent the use frequency of many outlets. However, as stated in lines 119-122 of the original manuscript, this system is operated in a manner to simulate the use of distal outlets, and we have now changed the wording to say “infrequently used” distal outlets. Future work could choose to operate the CMPRs with more frequent water changes, if desired. The text has been expanded to emphasize these points in lines 238-246:

‘The influent water to both CMPRs and stagnant pipes was initially seeded with backwash water from a granular activated carbon filter that had been in operation in a premise plumbing drinking water system for > 2 years to establish a mature microbial community. Water changes were conducted once weekly for the first 2 weeks of operation to facilitate colonization of the pipe surfaces. Water changes were then increased to twice weekly to better simulate an infrequent use pattern characteristic of distal outlets in large buildings, as a possible worst-case scenario for OPs control, until the end of the experiment (week 10). These periodic manual dump and fill water changes served to recreate turbulent intermittent flow and complete changeover of water, as occurs at infrequently used distal outlets.’

1-14 Lines 119-120, continued: Also, why did you change from once a week to twice a week partway through your experiment?

Response: The once weekly water changes during the first two weeks were selected to allow the microbes in the seed water to colonize the pipe walls and reduce potential for washout. Mention of the influent seeding has been moved to the beginning of this paragraph and the text has been changed in lines 240-244 in order to make this clearer:

‘Water changes were conducted once weekly for the first 2 weeks of operation to facilitate colonization of the pipe surfaces. Water changes were then increased to twice weekly to better simulate an infrequent use pattern characteristic of distal outlets in large buildings, as a possible worst-case scenario for OPs control, until the end of the experiment (week 10).’

1-15 Line 125 Could you include some basic information about the “base” (influent?) water that you used in your experiments (TOC, iron, etc.) in a table in the supplementary information? Also, you refer to this water as base water in some parts of the paper and influent water elsewhere. Choose one term and stick to it.

Response: Thank you. An additional table has been added to the SI (Table S. 2.) to provide this information. All references to ‘base water’ have been changed to ‘influent water.’

1-16 Line 128 No need to put the word seeded in quotes here or elsewhere.

Response: This change has been made throughout the text.

1-17 Line 129 Is the bacterial community in GAC backwash water likely to be representative of that found in a domestic plumbing system? If yes, could you explain why?

Response: Given that whole-house GAC filters are commonplace (i.e., 10% of homes in our experience), yes, we expect that the GAC backwash water with characteristically removed particles derived from the distribution system, is relevant to what enters a substantial fraction of domestic plumbing systems. Digging deeper into this comment, we would further expect that the microbial community composition would shift from the backwash water to the CMPR water, just as it would from a whole-house GAC unit to the residential plumbing. Still, the bulk of the microbes in premise plumbing would originate from the water supply, but various populations are enriched or diminished by the GAC filter and conditions thereafter. 

Figure S. 4. Directly addresses this comment, with an NMDS plot directly comparing the microbial community composition of the seed, influent, CMPR, and stagnant pipe waters. Figure S. 4. Is referred to in line 539. The shifts in microbial community composition are consistent with the above expectations, as we observed a difference between the microbial community structures of the influent, which was from a tap water outlet separate from the GAC filter, and the CMPR pipe effluent, which, as mentioned above, had been seeded with GAC backwash weeks before, and this difference was small. We note these details in lines 539-542:

‘Statistically, there was a difference in beta diversity between the influent water samples and bulk water samples from all CMPRs (Padonis=0.001 for both weighted and unweighted analysis); however, the magnitude of the difference was very small (R2=0.093; R2=0.068, unweighted and weighted, respectively).’

We also emphasize that the GAC filter used here has been used in a pilot-scale premise-plumbing system and flushed with tap water three times daily for the 2+ years of its operation and thus would provide a mature seed of influent microbes. The text has been updated in lines 238-240 to demonstrate that it had been used in a premise plumbing drinking water system;

‘The influent water to both CMPRs and stagnant pipes was initially seeded with backwash water from a granular activated carbon filter that had been in operation in a premise plumbing drinking water system for > 2 years to establish a mature microbial community.’

1-18 Lines 143-144 Why were these parameters only measured in week 2 and week 9? Why not measure them more frequently?

Response: The week 2 sampling was conducted to provide a profile of the CMPRs following their acclimation and during early stages of operation. Week 9 was selected as a representative time point when one would expect that the effects of pipe materials would be observable, based on prior studies referenced in line 279 

While more time points may have been interesting, they were not necessary to achieve the primary goal of demonstrating the ability of the CMPRs to distinguish effects of pipe material on water chemistry in a repeatable fashion, and as is already noted, we had sufficient replicates (n=16) to draw high statistical confidence in a single sampling event.

The justification for sampling at week 2 and week 9 is now provided in lines 277-279:

‘These timepoints were selected to profile the pipes following initial microbial colonization and at a later period when differences in biological water quality parameters would be expected based on pipe material [18, 19].’

This is briefly mentioned again in the caption for Table 1, now reading:

‘Table 1. Water quality parameters for replicate CMPRs during week 2 (acclimation, once weekly water change), week 9 (differentiation, twice weekly water change) or week 10 (qPCR data)’

1-19 Line 169 By circular motion do you mean that you swabbed the entire inner circumference of the pipe or that you took a swab from one part of the pipe (e.g. the bottom surface)?

Response: This phrase has been expanded to ‘and the entire inner surfaces (~13 cm2) of these endcaps were swabbed’ to better explain our method in lines 300-301. 

1-20 Line 194 Please describe the different blanks mentioned here.

Response: The included blanks are specified in lines 338: ‘…along with filter, swab, and extraction blanks…’. The list of blanks in line 307-308 has been updated to include the swab blank.

1-21 Lines 219-220 You could remove this sentence.

Response: We agree. The sentence has been removed.

1-22 Line 231 What are the expected pH trends for each type of pipe? Please describe briefly and provide citation(s). If these expected trends were observed in full scale domestic water heating systems it might be worth mentioning this in the final section of the results to support the idea that your CMPRs are comparable to real world systems.

Response: Corrosion on metallic pipes tends to raise pH, unless an insoluble metal hydroxide scale forms, in which case the pH will not change. Respiration is expected to lower the pH through the production of carbon dioxide, while the pH effect of other biological reactions will vary. Explanations of and references for these phenomena are now provided after the relevant results are discussed, in lines 379-395:

‘In week 2, PVC-copper pipes had a final pH of 7.66 ± 0.16 and PVC-iron pipes 7.72 ± 0.23, with a lower pH of 6.85 ± 0.06 in the PVC pipes (neach=18, p<0.0001 compared to each PVC-copper and PVC-iron CMPRs), presumably due to increased CO2 production via cellular growth and respiration [35, 36] in PVC CMPRs and corrosion in metallic CMPRs [37, 38]. An increase in bulk water pH from the service line to outlets has been previously observed in a residential building featuring copper plumbing [39].’ 

1-23 Lines 280 Why would individual iron pipes age at different rates if they are all new and all subjected to the same conditions?

Metal release from new iron and copper pipes has been shown to be highly variable, presumably due to small differences in the surface after manufacturing. We were funded by manufacturers to explore this issue and ended our work only showing that new copper pipes are often 50% variable from one another in terms of copper release. Unfortunately, we were never provided follow up funding to determine why. We are providing this unpublished report made to copper pipe manufacturers to the reviewer in supplemental documents.

This variability is almost never examined due to lack of replication in prior OPs studies (including our own), so we cannot cite statistics from prior research that proves this occurs. Supporting references are now provided in lines 439-440. Additionally, language was amended in lines 444-446 to both emphasize this expectation and highlight another instance in which the CMPRs replicated a known phenomenon in premise plumbing drinking water quality:

‘The variability in iron release was expected based on previous studies [46, 52, 53]. Between weeks 2 and 9, the CV decreased from 18.8% to 14.9% across CMPRs, consistent with iron release becoming more uniform as pipe-scale formed.’

1-24 Lines 283-286 Break this sentence into two sentences, one about copper and one about iron.

Response: This sentence was divided into two to better explain these two points in lines 471-474:

‘We hypothesize that the greater initial variability in PVC-copper pipes was due to cupric ions being released at different rates, resulting in different rates of microbial inactivation. In PVC-iron pipes, high DO consumption by corrosion may explain the initial variability of total cell counts (Fig S.2).’

1-25 Lines 284-285 Why would different individual copper pipes release different amounts of cupric ions if they were all new and all subjected to the same temperature conditions and water changeout regimes? This potentially points to a serious weakness in your approach or the execution of the study.

Response: This is the same question and response noted previously for iron. Copper pipes, even those from the same production batch, are highly variable in terms of copper release. It is speculated that this is due to variable oxide or carbon films on the pipe surface. Additional references for this point have been added in line 439-440:

‘Variability in copper release was fairly high, as is characteristic in new pipes [49, 50], with CVs of 32.2% during week 2 and 30.4% during week 9.’

 This variability, along with that observed in iron release from iron pipes, is one reason we judge that higher replication is recommended for future premise plumbing simulation studies and partly motivates the development of the CMPR to efficiently accommodates this need. We have attached the industry funded report, Influence of Drawing Lubricant on Copper Release from Copper Pipes During NSF 61 Testing, for additional documentation of very high variability in copper release from copper pipes. 

1-26 Lines 312-314 Add a citation to support this hypothesis.

Response: A reference has been added to line 509-510 (’… to a point where the biocidal activity had decreased [18]…’) which demonstrates copper pipes supporting similar cell growth to plastic pipes after sufficient aging. 

1-27 Lines 314-315 Have other studies found that iron pipes have lower DO than copper or PVC pipes?

Response: References have been added to line 506 that demonstrate iron corrosion’s consumption of DO, as well as a direct comparison of DO to that in copper pipes:

‘Iron also removes DO from the water through redox reactions, leaving less for use by cells for respiration and growth [44, 46].’

1-28 Lines 322-324 This should also be mentioned in the methods section.

Response: More details are offered on the sampling time points for total cell count in lines 291-292:

‘Bulk water total cell counts were taken during weeks 1, 2, 5, and 9, with an additional sampling of a random subset of CMPRs and pipes during week 6.’

1-29 Lines 324-325 The fact that new pipes were used does not indicate that pipe aging occurred, but the fact that you started with new pipes and your cell counts changed over time does suggest that pipe aging occurred. I suggest that you reword this sentence to clarify.

Response: This language has been changed to ‘the observed change in cell counts over the course of the experiment is likely due to aging of new pipes’ in lines 531-532.

1-30 Lines 325 Have other studies shown that pipe aging impacts cell counts? If yes, cite them here (you cited refs 37 and 38 to support a similar statement elsewhere in the paper). Depending on what kind of apparatus was used in these other studies, this could also be discussed in the final part of the results section.

Response: References demonstrating changes in total cell counts and HPCs with pipe age have been added to line 500-501, as the line referred to above is a figure caption:

‘Total cell counts were measured on a weekly basis to track expected changes in microbial numbers with pipe age (Fig 4A) [18, 54-56].’ 

1-31 Line 324 Why did you change your water changeover regime in week 2?

Response: This is addressed in our response to the comment for lines 119-120.

1-32 Line 330 Remove the word inherently.

Response: This word was removed.

1-33 Line 334 When you say “bulk water from the CMPRs”, do you mean that all of the data from all of the conditions (PVC, iron-PVC, copper) were grouped together and compared to the influent water? If yes, I would mention this in the text.

Response: This phrase has been expanded to ‘between the influent water samples and bulk water samples from all CMPRs’ for added clarity in line 540.

1-34 Lines 341-350 Has this relationship between microbial community and pipe material been observed in real world domestic hot water systems or in studies conducted with other lab scale apparatus? If yes, this would be an interesting thing to bring up in the final section of the results.

Response: References to studies that have made similar observations are now provided in lines 558-561:

‘Pipe type strongly influenced the composition of the microbial community that established in both the bulk water and biofilm, as has been observed in drinking water distribution systems [57], as well as other premise plumbing models such as pilot-scale rigs [58], simulated water heaters [59], and CDC biofilm reactors [60].’

This point is again emphasized in the final section, in lines 793-796:

‘These include chemical phenomena such as the observed differences in pH and DO, the variability and gradual reduction in copper and iron release, as well as biological differences such as total cells counts and microbial community composition in the bulk water and biofilm.’

1-35 Line 344 Remove the word however.

Response: This word has been removed.

1-36 Line 361 Did you actually observe sediments in your pipes?

Response: Sediments were regularly observed in the effluent of PVC-iron CMPRs. Line 594-595 has been changed to reflect this:

‘The greater variability in PVC-iron CMPRs may also be due to the influence of the sediments observed in the effluent of those pipes on the microbial ecology [62].’

1-37 Line 372 Why were all of the swabs taken from the PVC section of the pipes?

Response: This was necessary in order to ensure consistent and comparable recovery of biofilms, thus directly comparing the effects of the water chemistry on the biofilms at a consistent sampling surface. Depending on the study, researchers could choose to examine biofilms directly on the pipe surface of interest, but they would need to do so fully aware that corroded metal surfaces impede biofilm recovery and would need to design the experiment and the biofilm recovery system accordingly. This is now discussed in the manuscript in lines 303-305:

‘The PVC cap surfaces were selected for swabbing to ensure a consistent recovery of biofilm-associated microbes across conditions, as corrosion/deposition phenomena on the metal surfaces were anticipated to interfere with biofilm recovery.’

1-38 Lines 382-385 I don’t think that your DO findings, on their own, support all of this. I suggest that you tone down your language and bit and break this sentence into two sentences. The first sentence could begin with “this suggests that” rather than “thus” to emphasize that you are, to some degree at least, speculating about the causes underlying your DO results.

Response: This section has been changed to acknowledge that the explanation for these results in lines 623-626 is speculative:

‘This suggests that the convective mixing current subjected the mild steel pipe section to more corrosive conditions. Convective mixing likely increased the interactions of the bulk water with suspended solids and biofilm, increasing mass transfer and reactions with corrosion byproducts that consume DO’

1-39 Lines 386-389 This is an interesting point. Perhaps you could move it to the final section of the results to better support your assertions that the CMPRs can simulate real world conditions.

Response: This point, along with other real-world phenomena observed in the CMPRs, are now briefly emphasized in lines 791-807.

1-40 Line 418 Here and elsewhere in the paper you should write times, rather than x.

Response: These changes have been made. ‘1x’ and ‘2x’ in reference to water change frequency have been changed to ‘once’ and ‘twice.’

1-41 Lines 459-482 This section is, in my opinion, the heart of the paper, but it contains a number of unsupported or poorly supported claims by the authors and as a result it is ultimately unconvincing. For example, reference 10 was used to support the statement that pilot-scale plumbing rigs had higher inherent variability than your CMPRs, but when I quickly reviewed that paper, I saw no mention of high variability. In general, this section would be more effective if you focused on comparing your actual water quality findings (e.g. DO, cell counts, etc.) to the results of studies that examined full scale domestic hot water systems or that used CDC reactors, pilot pipe rigs, etc. This would show that your CMPRs were comparable or superior to existing methods. There is some of this in various parts of the results section (lines 338-340, lines 386-389, etc.) and this content could be moved to this final section to

better support your claims.

Response: As the reviewer mentioned, we agree that this would be extremely valuable, but as the reviewer also acknowledges, it is not feasible in this study. 

To our knowledge such a comparison has never been made for any of the other premise plumbing simulation systems, including CDC reactors, pilot rigs, etc. We now have added text giving a list of reasons why such a comparison would be difficult-- mainly because premise plumbing itself is essentially infinitely variable in flow patterns, temperature, and disinfectant profiles. 

Our intention is that the heart of the paper is the introduction and initial evaluation of the CMPR as a new premise plumbing simulation system and a frank first-time discussion of some practical strengths and weaknesses that could help other researchers make better decisions regarding which test approaches to apply in their future work. In forthcoming research projects, we will further evaluate the CMPRs for addressing specific research questions. Note that a comprehensive research effort comparing the efficacy of lab simulations to replicate problematic niches in premise plumbing would be a massive undertaking and one that would be difficult to justify and secure appropriate funding to achieve. 

In this study, we do provide substantial evidence that the CMPRs would be at least as comparable as currently available laboratory simulations, while also being frank in acknowledging advantages, disadvantages, and unknowns. 

Exemplary statements and changes in the revised manuscript in support of our response to Comment 1-41 include the following:

• Lines 406-411: ‘DO tended to be lower in the PVC pipes than in the PVC-copper pipes during week 2 (ncopper=9, nPVC=6, p=0.0018), although there was no difference between PVC-copper and PVC pipes following 9 weeks of acclimation. This is likely due to the aging of PVC-copper pipes, resulting in increased cell growth due to the decreased release of antimicrobial copper as the pipes aged [19, 47], which in turn allowed more cellular respiration to consume DO, as was observed earlier in the experiment for the PVC pipes.’

• Lines 443-446: ‘Average total iron decreased from 24.9 ± 4.68 mg/L at week 2 to 10.9 ± 1.62 mg/L at week 9, consistent with scale formation [51]. The variability in iron release was expected based on previous studies [46, 52, 53].’

• Lines 505-506: ‘Iron also removes DO from the water through redox reactions, leaving less for use by cells for respiration and growth [44, 46].’

• Lines 558-561: ‘Pipe type strongly influenced the composition of the microbial community that established in both the bulk water and biofilm, as has been observed in drinking water distribution systems [57], as well as other premise plumbing models such as pilot-scale rigs [58], simulated water heaters [59], and CDC biofilm reactors [60].’

• Lines 628-630: ‘The effect of lower DO in the CMPRs is desired in this case, as DO depletion is a common phenomenon in “real-world” water systems where galvanized iron pipes have corroded following the loss of the protective zinc layer [46].’

1-42 Lines 459-482, continued: The cost information is interesting but ultimately costs will differ from one jurisdiction to another and over time, so I wouldn’t make it the main focus here.

Response: We agree and use the words “estimated” in line 734 (‘Pilot-scale premise plumbing rigs constructed in prior studies were estimated to cost…’) and “representative” in the bottom of Table 2 (‘Representative Cost, USD per replicate’). Additionally, we acknowledge they will differ in the updated caption for Table 2:

‘Table 2. Comparison of the CMPR to alternative reactors for simulating premise plumbing based upon available pricing and other information for required equipment components and personal knowledge and experience of the authors. Y = Yes, N = No, D = Difficult/Expensive’

And in line 732, we preface this paragraph with the phrase, ‘while costs of any system will vary with time and location.’

1-43 Lines 459-482, continued: Also, assuming that stagnant pipe reactors like those used in this study are widely used to simulate domestic hot water systems, they should be included in Table 2 and discussed in this section.

Response: We did not mean to imply that this approach is superior to other premise plumbing models, or that it replicates every niche in premise plumbing. This novel approach does recreate a plumbing niche very important to OPs. We have emphasized research demonstrating some specific, real-world phenomena that our system replicates. 

However, to quantitatively compare water quality findings from CMPRs to those of other models, simultaneous studies using the same source water would need to be performed as discussed earlier. We agree that the best way would be to directly compare several systems to a real-world building all using the same water source. To our knowledge, no prior studies have statistically compared one reactor approach to another, much less to results of a full-scale building. An additional paragraph to elucidate this position has been added in lines 752-764:

‘The ultimate basis of comparison, however, should be the ability of these systems to replicate real-world premise plumbing conditions. Here, we have shown that the CMPRs replicate phenomena observed in some important real-world settings. Unfortunately, it is not possible to compare our water quality findings to those of other full-scale systems because every water has a unique chemistry and every sampling location in a building is unique in terms of water use patterns, temperature and disinfectant time profiles. Such a comparison would be useful but it would be a major undertaking in terms of personnel and resource requirements, but would be highly valuable because it is important to obtain a better understanding of the real-world strengths and limitations of each laboratory simulation method. A particular strength of CMPRs is an ability to be modified to accommodate a wide range of replicates, pipe conditions, temperature regimes, and adjustments to influent water chemistry. We judge that CMPRs are a useful, novel and cost-effective test method that can help to better understand a wide variety of microbiological and physicochemical premise plumbing research issues.’

Furthermore, the bullet point beginning on line 491 of the original manuscript has been moved to the end of that list and expanded to place more emphasis water quality phenomena CMPRs replicate. Lines 791-807 now read:

‘CMPRs were able to induce and isolate certain effects of key test variables, e.g., pipe material effect on water chemistry and microbiology, that have been observed as trends in other premise plumbing simulations and full-scale buildings. These include chemical phenomena such as the observed differences in pH and DO, the variability and gradual reduction in copper and iron release, as well as biological differences such as total cells counts and microbial community composition in the bulk water and biofilm. This enables future testing to explore impacts of a wider range of chemistries and experimental conditions while rigorously testing for statistical confidence between true replicate reactors.’

Additionally, earlier edits emphasize comparisons between our water quality findings to those from other pilot system and full-scale building studies. See responses to Comments 1-22, 1-23, 1-25, 1-27, 1-30, & 1-34.

Our emphasis on the differences between stagnant pipes and CMPRs is intended to demonstrate the possible importance of convective mixing in facilitating the interactions between the bulk water, biofilm, and pipe wall that would be expected to occur in premise plumbing, as well as their role in achieving conditions suitable for microbial growth for the study of OPs. 

1-44 Line 460 You didn’t measure nitrogen or discuss it in the rest of the paper, so I suggest that you omit it from this list.

Response: Nitrogen has been removed in line 718.

1-45 Line 461 This is the first time that your base / influent water is described as oligotrophic. This term may not be familiar to your readers, so I think it would be more effective to briefly describe the important characteristics of your water matrix (low TOC, moderate pH, dechlorinated, etc.)

Response: We did not mean to imply that the water used in this study is more oligotrophic than other drinking waters are expected to be. ‘Oligotrophic drinking water’ has been changed to ‘the low-nutrient environment of drinking water’ in lines 718-719. A table has been added to the SI to provide information regarding the influent water.

1-46 Line 476 Are mechanical failures truly unavoidable with the CDC reactors? Couldn’t the UV light, recirculation pump, heater, etc. in the CMPR set-up fail as well?

Response: Our experience in running multiple parallel reactors, is that mechanical failures and snafus are unavoidable, and they occur randomly to all reactors at some rate over the months needed to do a representative study, and that as a result none of the reactors are truly replicated. Moveover, the beauty of the CMPR, is that when (not if) a failure occurs, it is not only less frequent because there is only one of everything, but it happens to all reactors equally. So replication between reactors is maintained, even during the failures.

We did not intend to imply that individual components of the CMPR system are less likely to fail than those found in other reactors. Rather, that the need for multiple peristaltic pumps and stir plates for each pipe material when using CDC biofilm reactors introduces many more chances for an individual component to fail. This sentence has been updated in lines 743-747 as follows: 

‘This makes testing multiple pipe conditions in parallel expensive and often cost prohibitive, as an individual reactor with full controls will be needed for each pipe type tested, while also introducing a greater chance of mechanical failure of at least one component. Moreover, when a failure of a component inevitably occurs in a CMPR, true replication is maintained because it will affect every pipe equally.’

1-47 Line 489 Some of your results, especially from week 2, do not support this conclusion.

Response: The phrase ‘beyond that which is inherent to new pipe materials’ has been added to line 783 in order to better specify the observed results. This, along with the above changes to address the natural variability of new pipe sections, should indicate more precisely what was found.

1-48 Figure 1 What does the checked box represent? Also, it might be nice to include a schematic showing the stagnant pipe apparatus for comparison.

Response: The figure has been revised so that the manuscript’s text clarifies that the checked box is an interior view of convective mixing in the pipe. In-text references to the schematic portion of Figure 1 (Fig 1A) and the interior view portion (Fig 1B) are now in lines 132, 133, 206 and 207. Changes to the figure caption referencing these separate panels are made in lines 142-148:

‘Fig 1. Schematic of convective mixing pipe reactors (CMPRs). (A) Capped ends of PVC-copper, PVC-iron, and PVC pipes are submerged at 60° in a hot water bath to simulate hot water recirculation, with plugged, accessible ends exposed to room temperature to simulate a stagnant distal outlet. Fifty-four pipes, configured into 6 rows of 9 pipes, were operated for this study. In-line ultraviolet light is incorporated for disinfection of water bath for secondary containment/disinfection of pathogens in case of leak to support BSL2 level experiments.’

We feel a visual representation of the stagnant pipes is not necessary, as they are much simpler to construct than the CMPRs and lines 220-221 of the manuscript provide sufficient detail for their complete replication. Additional details (‘…triplicates of each pipe were placed at a 60° angle against a shelf …’) have been provided in lines 220-221 to describe our particular configuration and facilitate reader’s understanding of the set-up.

1-49 Figure 2 I must admit to initially being totally confused by the letters on these plots because the way that you’ve indicated statistical differences is unfamiliar to me and, in my opinion, counterintuitive. I’m used to statistical differences being indicated by different letters, rather than by the same letters. So, for example, if condition 1 and condition 2 were statistically different, they would be labeled A and B, but if they were indistinguishable, they’d both be labeled A because they would be in the same group. This is in line with the outputs of all of the statistical programs that I’ve ever worked with. I think that the approach that I’ve described here is likely to be more familiar and intuitive to readers, so I suggest that you adopt it here.

Also, why do some plots only have capital letters and some only have lower case letters? Weren’t the same statistical tests run on all four datasets?

Response: Figures 2 and 3 have been edited so that letters indicate grouping rather than differences. The captions for these figures have been updated accordingly. 

1-50 Figure 4 It would be interesting to see DO, cell counts, and iron plotted together (or on three faceted plots in a single figure) to emphasize how they influence one another.

Response: We have created a new plot and added this figure to the SI (Figure S. 2), comparing DO, cell counts, and iron data from weeks 2 and 9. Reference to this figure is provided in line 474. While visually the three parameters appear to change in sync as the pipes age, we were unable to identify a statistically-significant correlation between cell counts and either DO or iron concentration. We suspect that this is because of the natural variability in these parameters, as well as the fact that DO and iron only have two sampling points as mentioned earlier. 

1-51 Table 2 This table is interesting but the authors don’t support their assumptions with any citations and it isn’t always clear how they came to their conclusions about different factors. For example, why wouldn’t it be possible create a standardized protocol for a pilot-scale pipe rig? Why would the CMPRs cost less to operate than a CDC reactor, especially seeing as the CMPR apparatus includes a UV lamp and heater, both of which would consume electricity? Why would it be more difficult or expensive to ensure biosafety protections for CDC biofilm reactors? Also, the authors should specify that their costing is USD.

Response: The assessment summarized in Table 2 is “based upon available pricing and other information for required equipment components and personal knowledge and experience of the authors,” which is now explicitly stated in the title of Table 2 in lines 724-726. A full explanation of each value in this table would form a substantial review article on its own and is better suited for a study that directly compares these systems. On the issue of cost, for example, the CMPR only needs one pump, UV lamp and heater, whereas every replicate CDC reactor requires one pump and one stir plate. The scaling of replicative costs are linear in the case of the CDC reactor. Thus, we make the case that the assumptions built into Table 2 are reasonable based on available information, while also making it clear that more precise comparison would require a separate study. USD and other caveats have been added to the bottom line of Table 2.

---

## [Decision Letter · Decision Letter 1]

17 Aug 2020

Replicable simulation of distal hot water premise plumbing using convectively-mixed pipe reactors

PONE-D-20-13252R1

Dear Dr. Edwards,

We’re pleased to inform you that your manuscript has been judged scientifically suitable for publication and will be formally accepted for publication once it meets all outstanding technical requirements.

Kind regards,

Zhi Zhou, Ph.D.

Academic Editor

PLOS ONE

Additional Editor Comments (optional):

Reviewers' comments:

Reviewer's Responses to Questions

**Comments to the Author**

1. If the authors have adequately addressed your comments raised in a previous round of review and you feel that this manuscript is now acceptable for publication, you may indicate that here to bypass the “Comments to the Author” section, enter your conflict of interest statement in the “Confidential to Editor” section, and submit your "Accept" recommendation.

Reviewer #1: All comments have been addressed

2. Is the manuscript technically sound, and do the data support the conclusions?

Reviewer #1: Yes

3. Has the statistical analysis been performed appropriately and rigorously? 

Reviewer #1: Yes

4. Have the authors made all data underlying the findings in their manuscript fully available?

Reviewer #1: Yes

5. Is the manuscript presented in an intelligible fashion and written in standard English?

Reviewer #1: Yes

6. Review Comments to the Author

Reviewer #1: Thanks for addressing all of my comments. It's unfortunate that it wasn't feasible to compare your results to full scale systems for this paper, but I suppose that just means that there is still some interesting work to be done in this space.

7. PLOS authors have the option to publish the peer review history of their article (what does this mean?). If published, this will include your full peer review and any attached files.

Reviewer #1: **Yes: **Stephanie Gora

---

## [Editor Report · Acceptance letter]

20 Aug 2020

PONE-D-20-13252R1 

Replicable simulation of distal hot water premise plumbing using convectively-mixed pipe reactors 

Dear Dr. Edwards:

I'm pleased to inform you that your manuscript has been deemed suitable for publication in PLOS ONE. Congratulations! Your manuscript is now with our production department. 

Kind regards, 

on behalf of

Dr. Zhi Zhou 

Academic Editor

PLOS ONE